

**The Atmospheric Oxidizing Capacity in China: Part 1. Roles of different photochemical processes**

Jianing Dai[a], Guy P. Brasseur[a,e,f], Mihalis Vrekoussis[b,g,h], Maria Kanakidou [b,d], Kun Qu[b], Yijuan Zhang[b], Hongliang Zhang[c], Tao Wang[f]

[a] Environmental Modelling Group, Max Planck Institute for Meteorology, Hamburg, 20146, Germany

[b] Institute of Environmental Physics (IUP), University of Bremen, Bremen, 28359, Germany

[c] Department of Environmental Science and Engineering, Fudan University, 200433, China

[d] Environmental Chemical Processes Laboratory, Department of Chemistry, University of Crete, Heraklion, 71003, Greece

[e] National Center for Atmospheric Research, Boulder, Colorado, 80307, USA

[f] Department of Civil and Environmental Engineering, The Hong Kong Polytechnic University, Hong Kong, China

[g] Center of Marine Environmental Sciences (MARUM), University of Bremen, Germany

[h] Climate and Atmosphere Research Center (CARE-C), The Cyprus Institute, Nicosia, Cyprus

*Correspondence to*: Guy P. Brasseur (guy.brasseur@mpimet.mpg.de)





**Abstract**

The atmospheric oxidation capacity ($AOC$) characterizes the ability of the atmosphere to scavenge air pollutants. However, it is not well understood in China, where anthropogenic emissions have changed dramatically in the past decade. A detailed analysis of different parameters that determine the $AOC$ in China is presented on the basis of numerical simulations performed with the regional chemical-meteorological model WRF-Chem. The model results, with the aerosol effects of extinction and heterogeneous processes taken into account, show that the presence of aerosols leads to a decrease in surface ozone of approximately 8-10 ppbv in $NO_x$-limited rural areas and an increase of 5-10 ppbv in VOC-limited urban areas. The ozone reduction in $NO_x$-sensitive regions is due to the combined effect of nitrogen dioxide and peroxy radical uptake on particles and of the light extinction by aerosols, which affects the photodissociation rates. The ozone increase in VOC-sensitive areas is attributed to the uptake of $NO_2$ by aerosols, which is offset by the reduced ozone formation associated with $HO_2$ uptake and with the aerosol extinction. Our study concludes that more than 90% of the daytime $AOC$ is due to the reaction of the hydroxyl radical with VOCs and carbon monoxide. In urban areas, during summertime, the main contributions to daytime $AOC$ are the reactions of OH with alkene (30-50%), oxidized volatile organic compounds (OVOCs) (33-45%), and carbon monoxide (20-45%). In rural areas, the largest contribution results from the reaction of OH with alkenes (60%). Nocturnal $AOC$ is dominantly attributed to the nitrate radical (50-70%). Our results shed light on the contribution of aerosol-related $NO_x$ loss and the high reactivity of alkenes for photochemical pollution. With the reduction of aerosols and anthropogenic ozone precursors, the chemistry of nitrogen and temperature-sensitive VOCs will become increasingly important. More attention needs to be paid to the role of photodegradable OVOCs and nocturnal oxidants in the formation of secondary pollutants.

Keywords: $O_3$, Atmospheric Oxidation Capacity, Heterogeneous Reactions.









## 1. **Introduction**

With the drastic actions initiated by the Chinese authorities to improve air quality, specifically to reduce the emissions of primary pollutants, including nitrogen oxides ($NO_x$), volatile organic compounds (VOCs), carbon monoxide (CO), sulfur dioxide ($SO_2$), and the concentration of particulate matter (PM) suspended in the atmosphere, the level of several secondary pollutants including near-surface ozone ($O_3$) has increased significantly between years 2013 and 2019, most notably in the

North China Plain (e.g., Lu et al., 2018; Liu and Wang, 2020., Wang et al., 2020). Several papers have documented the observed trends of $O_3$ in China (Lu et al., 2018; Wang et al., 2022). In some cases, studies have provided some explanation for the cause of these trends (Li et al., 2019a., Liu and Wang et al., 2020), specifically in the most polluted areas, or have proposed some mitigation strategies (Li et al., 2019b). Among the formulated hypotheses to explain these trends, the most

feasible explanation is the reduction in the level of $NO_x$ in the polluted planetary boundary layer (PBL) with a related reduction in the rate at which $O_3$ is titrated by nitric oxide (NO) in VOC-limited areas. Another potential cause for the observed $O_3$ increase is the reduction in the atmospheric aerosol burden and hence in the rate at which peroxy radicals ($HO_2$ and $RO_2$) that contribute to $O_3$ formation are removed by heterogeneous processes (Li et al., 2019a; Liu and Wang, 2020).


Alleviating $O_3$ pollution requires a quantitative understanding of the different chemical processes that contribute to the photochemical formation and destruction of secondary species. It also requires detailed investigation of the budget of fast-reacting radicals that are directly involved in photochemical oxidation processes. Recent observational studies have documented and analyzed

the evolution of several reactive species, such as OH, $HO_2$ and $RO_2$ radicals and $O_3$. On the basis of observations, generally at a single location with routine measurements lasting for several years (Liu et al., 2022, Tan et al., 2019., Zhu et al., 2021., Wang et al., 2022). The sensitivity of ozone, particulate matter and oxidative processes to precursor emissions has also been studied in previous work (Liu et al., 2010; Xing et al., 2017) under highly polluted conditions. Such studies need to

be repeated for current conditions characterized by reduced pollutant emissions and aerosol loading.

The purpose of the present study is to provide a quantitative estimate of the different factors that affect the oxidation capacity of the atmosphere in the entire geographical area covered by China.

The concept of atmospheric oxidation capacity (*AOC*) has been introduced several decades ago (e.g., Thompson, 1992; Prinn, 2003) to highlight the existence of self-cleansing processes in the atmosphere. These processes allow the removal of most primary pollutants, including methane ($CH_4$), non-methane hydrocarbons (NMHCs), CO, $NO_x$, $SO_2$, as well as the formation of secondary species, including $O_3$, particulate nitrate ($NO_3^-$), sulfate ($SO_4^{2-}$), and secondary organic aerosols

(SOA). The oxidation capacity is a measure of the ability of the atmosphere to destroy primary





species emitted at the Earth's surface. It is directly linked to the presence of highly reactive radicals, including OH and nitrate radical ($NO_3$). It is therefore influenced by processes such as photolysis generated by solar radiation, temperature, emission, scavenging processes, atmospheric transport, and other meteorological factors. It is characterized by different factors, including the atmospheric production rate of $RO_x$ radicals (with $RO_x$ defined as $OH + HO_2 + RO_2$, where R represents an organic chain) and the OH reactivity.

Based on model simulation, the analysis presented in this paper (Part 1) assesses the relative importance of different photochemical processes that contribute to the formation and destruction of near-surface $RO_x$ and $O_3$ in different chemical environments encountered in China. A companion paper (Part 2; Dai et al., 2023) focus on the role of emission changes on the photo-oxidative species and parameters. This paper is structured as follows. Section 2 first provides some theoretical considerations on which our analysis is based. The adopted regional chemical-meteorological model, described in Sect. 3, is driven by reanalyzed meteorology for the year 2018 and by regional surface emissions that account for the contribution of different sectors. The analysis of the model simulations performed for different conditions is presented in Sect. 4-6. Specifically, the budget of oxidants ($RO_x$ and $O_3$) is discussed in Sect. 4. The effect of heterogeneous chemical processes in the presence of aerosols is addressed in Sect. 5. Section 6 provides a quantitative estimate of the different indicators that describe the oxidation capacity of the atmosphere. A summary of the principal findings is provided in Sect. 7. Additional information, including the validation of the model simulations, can be found in the Supplementary Information.

## 2. Theoretical considerations

### 2.1. $RO_x$ radicals

As shown in the pioneering paper by Levy (1971), the fate of many primary atmospheric species ($CO$, $CH_4$, and NMHCs) and the formation of secondary species, including tropospheric $O_3$, are associated with cycling chain reactions involving OH, $HO_2$, and $RO_2$. In the theoretical description of the key chemical processes presented here, we refer to the simplified reaction scheme listed in Table 1, but the chemical mechanism adopted in our model is considerably more detailed.

The production of $RO_x$ radicals in the troposphere results primarily from the photolysis of $O_3$ (reactions R1 and R8, see Table 1), of nitrous acid (HONO) (reaction R5), and of different oxygenated volatile organic compounds (OVOCs) such as formaldehyde (HCHO) (reaction R3), larger aldehydes, acetone, etc. (reaction R4). The ozonolysis of alkenes (Alk) (reaction R11) is an additional source of $RO_x$, which is believed to play a relatively minor role. When considering the sources due to the photolysis of OVOCs, we single out formaldehyde due to the major contribution of this species to $RO_x$ production. OVOCs stand, therefore, for the remaining non-HCHO OVOCs. Thus, we express the production rate of $RO_x$ as



$P(RO_x) = 2\,k_8\,[O(^1D)]\,[H_2O] + J_{HONO}\,[HONO] + J_{HCHO}[HCHO] + \Sigma_i\,r_i\,J_i\,[OVOC_i]$
$+ \Sigma_i\,s_i\,k_{11,i}\,[Alk_i]\,[O_3],$
(1)

Here and in other expressions below, factors $k_i$ represent the reaction coefficients for the reaction $i$ in Table 1, and $J$ is the photolysis frequency for the chemical species under consideration. The brackets stand for the number densities of species generally expressed in molecules or radicals per $cm^3$. Coefficient $r_i$ represents the number of $RO_x$ produced by the photolysis of each OVOCs species and $s_i$ is the number of $RO_x$ produced by each alkene ozonolysis reaction. These coefficients are specific to a reaction involving the photolysis of OVOCs and the ozonolysis of the alkenes. A more explicit form for the two last terms in the above expression depends on the adopted chemical mechanism. The $RO_x$ source term provides an estimate for the availability of radicals that initialize photooxidation processes in the troposphere.

The destruction of $RO_x$ radicals results from the termination reactions R15-18 between different $RO_x$ radicals ($L_H$), reactions R22-24 between $RO_x$ radicals and nitric oxide ($L_N$) and the heterogeneous uptake of $HO_2$ (reaction R28) on aerosol surfaces ($L_{het}$). Thus, the total destruction rate of $RO_x$ can be expressed as

$D(RO_x) = L_H + L_N + L_{het},$
(2)

with

$L_H = \{\, 2\,k_{15}\,[OH] + 2\,k_{16}\,[HO_2] + 2\,\Sigma_i\,k_{17}\,[RO_{2,i}]\}\,[HO_2] + 2\,\Sigma_{i,j}\,k_{18,\,i,\,j}\,[RO_{2,i}]\,[RO_{2,j}]$

$L_N = \Sigma_i\,k_{22,\,i}\,[RO_{2,i}]\,[NO] + k_{23}\,[OH]\,[NO_2] + k_{24}\,[OH]\,[NO]$

$L_{het} = k_{28}\,[HO_2]$

In the above expressions, we assume that, near the surface, the $RO_x$ termination reactions between $HO_2$ and $NO_2$ (that produce nitrous acid, $HO_2NO_2$) and between acetyl peroxy radicals and $NO_2$ (that produce peroxyacetyl nitrate, PAN) are balanced by the regeneration of $RO_x$ resulting from the photolysis of $HO_2NO_2$ and the thermal decomposition of PAN (equilibrium conditions), respectively. Therefore, the related reaction rates do not appear explicitly in the above expressions.

2.2. Odd Oxygen





In the troposphere, odd oxygen ($O_x = O_3 + NO_2$) is produced through complex recurrent radical reaction chains involving the oxidation of hydrocarbons in the presence of $NO_x$. The $O_3$ molecule is formed by the rapid photolysis of $NO_2$ (reaction R2) followed by the recombination of atomic oxygen (reaction R7). Reaction R2 is balanced by reactions R19 and R20, which reproduce $NO_2$.
A production of odd oxygen occurs only if NO is converted to $NO_2$ without consuming $O_3$, i.e., by reactions R20 and R21 with a peroxy radical ($HO_2$, $CH_3O_2$, and other higher-order organic radicals) provided by the oxidation of methane, other hydrocarbons (HC), and by carbon monoxide (reactions R9, R10 and R12). The resulting production rate of odd oxygen can be expressed with a good approximation by


$$P(O_x) = k_{20} [HO_2] [NO] + \Sigma_i\, k_{21,i} [RO_2]_i [NO],$$
(3)

This equation highlights the nonlinear nature of $P(O_x)$ since the concentration of nitric oxide and
peroxy radicals are dependent on each other.

The photochemical destruction of $O_x$ results from several processes, including the photolysis of $O_3$ (reaction R1) followed by the reaction R8 between the electronically excited oxygen atom $O(^1D)$ and water vapor ($H_2O$). Other $O_x$ loss mechanisms involve the reactions of ozone with OH
(reaction R13), $HO_2$ (reaction R14), and different alkenes (Alk) (R11). In the presence of $NO_x$, an additional loss mechanism is provided by the titration of $O_3$ by NO (R19), followed by the conversion of $NO_2$ to nitric acid ($HNO_3$; R23). The total destruction rate of $O_x$ is therefore expressed as

$$D(O_x) = k_8 [O^1D][H_2O] + \{\Sigma_i\, k_{11} [Alk_i] + k_{13} [OH] + k_{14} [HO_2] + k_{19} [NO]\}[O_3] + k_{23}[OH][NO_2],$$
(4)

The dominant pathways leading to the formation and destruction rates of $O_x$ and hydroxyl radicals vary according to chemical environments. Under relatively clean conditions with low levels of
$NO_x$, the production of odd oxygen, as provided by reactions R20 and R21, is limited by the availability of $NO_x$, while the loss of the $RO_x$ radicals is dominated by the peroxy-radical self-reaction R16 that leads to the formation of hydrogen peroxide ($H_2O_2$) (Song et al., 2021). Under these conditions, a reduction in the emissions of $NO_x$ tends to reduce the ground-level $O_3$ concentration.

In polluted areas, including many urban centers, the level of $NO_x$ is so high that saturation conditions prevail. In this environment, the formation of $O_3$ is determined by the availability of VOCs, and the loss of $RO_x$ is dominated by reaction R23 between $NO_2$ and OH, which produces $HNO_3$. In this case, a reduction in $NO_x$ tends to increase the concentration of $O_3$, while a reduction in VOCs is expected to reduce its near-surface abundance (Wang et al., 2022). Furthermore, since
highly polluted environments are generally characterized by elevated aerosol loads, the effect of





heterogeneous processes on the abundance of reactive species becomes particularly important. Among these reactions, we consider more specifically the uptake of $HO_2$, $NO_2$, $NO_3$, and $N_2O_5$ on the atmospheric aerosol surfaces (reactions R28-31). The heterogeneous destruction of peroxy radicals on the surface of aerosols tends to inhibit the formation of $O_3$ by reducing the rate of reactions R20 and R21. Under high levels of aerosols, this reaction may become important for $O_3$ formation, which has led Ivatt et al. (2022) to define a third $O_3$ sensitivity regime called the aerosol-inhibited photochemical $O_3$ regime. The uptake of $NO_2$ leads to the formation of HONO, whose photolysis represents a significant source of OH. It also forms NO, which reacts with $HO_2$ and $RO_2$ to produce $O_3$. In short, heterogeneous processes may either favor or inhibit the formation of odd oxygen in polluted areas.

The lifetime of $O_x$ in the PBL is sufficiently long (one or two days) that additional processes besides photochemical production and destruction need to be taken into consideration. Among them is the additional loss of $O_3$ and $NO_2$ resulting from dry deposition on the vegetation. With a deposition velocity of about 1 cm s$^{-1}$ (Wesely et al., 2000), the corresponding odd oxygen loss rate in the boundary layer is close to 1 ppbv h$^{-1}$, if one assumes that the depth of the mixing layer is in the order of 1 km. In addition, vertical mixing in the convective PBL and advective horizontal transport tend to disperse locally produced $O_3$ and balance its net production, for example, in urban centers.

2.3. Formulation of aerosol uptake

As stated above, heterogeneous chemical reactions can substantially influence the concentrations of $RO_x$ radicals and $O_3$. The reactions under consideration in our analysis are reactions R28 to R31, listed in Table 1. The first-order reaction rate constant on aerosols $k_a$ [s$^{-1}$] for species $i$ associated with these reactions is expressed by (Schwartz (1986)):

$$k_{a,i} = A_a \left[ \frac{a}{D} + \frac{4}{\gamma_{a,i} v_i} \right]^{-1},$$
(5)

where $A_a$ [cm$^2$ cm$^{-3}$] is the aerosol surface area density, $a$ [cm] is the mean radius of the particles, $D$ [0.247 cm$^2$ s$^{-1}$] is the gas-phase diffusion coefficient (Mozurkewich et al., 1987), $\gamma_a$ is the dimensionless reaction-dependent uptake coefficient for species $i$, and $v_i$ [cm s$^{-1}$] is the mean thermal velocity of species $i$ given as a function of temperature $T$ [K] and molecular mass $m_i$ by

$$v_i = \left[ \frac{8 k_b T}{\pi m_i} \right]^{\frac{1}{2}},$$
(6)

with $k_b$ (1.38 $\times 10^{-23}$ J K$^{-1}$) being the Boltzmann constant.




The chemical substances produced as a result of the $HO_2$ uptake onto aerosol surfaces are not clearly established and could be $H_2O$ or $H_2O_2$ (Song et al., 2021). Here, to determine the maximum effect of this reaction, we assume that the $HO_2$ uptake onto aerosols (reaction R28) represents a terminal reaction of the hydrogen radical chain. Hence, water molecules rather than peroxide mol-

ecules are assumed to be formed. The corresponding uptake coefficient ($\gamma_{HO2}$) is chosen to be 0.1 in this study, a value lower by a factor of 2 than used in some earlier model studies ($\gamma_{HO2}$ = 0.2; Tie et al., 2001, 2005; Martin et al., 2003; Liu and Wang, 2020; Ivatt et al. 2022), but consistent with the conclusions reached by Gaubert et al. (2020) from their model simulations. These lower values are also adopted by Yang et al. (2022) and are consistent with the measurements of Lakey

et al. (2015), Tan et al. (2020), and Song et al. (2020). Specifically, on the basis of observations made in the Beijing-Tianjin-Hebei area during the summer of 2014, Song et al. (2020) conclude that the best fit for the value of $\gamma_{HO2}$ is a value of 0.116 ± 0.086, which is close to the value adopted in the present study.

The heterogeneous uptake of $N_2O_5$ by aerosol particles leads to the formation of nitric acid molecules (reaction R29). In the present study, we neglect the possible formation of $ClNO_2$ followed by its photolysis into Cl and $NO_2$. This process associated with the presence of chloride ions in the bulk of the particles is a source of additional radicals and hence could have an influence on $O_3$ (Thornton et al., 2010; Dai et al., 2020). Here, for the uptake of $N_2O_5$, we adopt the first-order rate

constant as expressed by Bertram et al. (2009) and modified by Yu et al. (2020) with the surface concentrations of the particle chloride and nitrate ions taken from the MOSAIC estimates. This parameterization has been used in simulating the concentration of $N_2O_5$ in several Chinese sites (Yu et al., 2020), and the simulated levels are in good agreement with the observed $N_2O_5$ values (Dai et al., 2020).


The rate of heterogeneous conversion of $NO_3$ (reaction R30) is calculated by Eq. (5) with a value of the uptake coefficient equal to $10^{-3}$ (Jacob, 2000; Xue et al., 2014; Liu and Wang et al., 2020).

Finally, the heterogeneous uptake of $NO_2$ on aerosol surfaces leads to the production of HONO

and $HNO_3$ (reaction R31), and, as HONO is rapidly photolyzed after sunrise, this heterogeneous process represents a source of OH radicals. The process also converts $NO_2$ into NO. Here, according to Zhang et al. (2021), we express the first order rate constant by Eq. (5) with a value of the uptake coefficient equal to $8\times10^{-6}$ during nighttime and to $1\times10^{-3} \times (J/J_{max})$ during daytime (Li et al., 2010; Czader et al., 2012; Fu et al., 2019), with $J$ representing the light intensity [W m$^{-2}$] and

$J_{max}$ [W m$^{-2}$] the peak value of light intensity (chosen to be 400 W m$^{-2}$ in this study).

### 2.4. Other HONO sources



For the particular heterogeneous reaction involving $NO_2$, which leads to the formation of nitrous acid, we also consider the additional contribution of the uptake on flat surfaces, specifically on bare soils, including asphalt in urban areas. This effect is believed to play a significant role particularly in urban areas (Zhang et al., 2016; Li et al., 2018). Zhang et al. (2021) claim that the measured vertical nighttime profile of this species suggests that the dominant formation nighttime mechanism of HONO results from the heterogeneous conversion of $NO_2$ on the ground. For this process, which is only crudely represented here, we assume that the first-order rate constant $k_g$ [s$^{-1}$] for this process is given by Liu et al. (2019) for nighttime conditions

$$k_g = \frac{1}{8} \gamma_g v_{NO_2} A_g,$$
(7)

where $A_g$ is the surface area density over the bare soil and urban surfaces; $\gamma_g$ is the uptake coefficient on the ground. Here, according to Zhang et al. (2021), we express a value of the uptake coefficient equal to $4 \times 10^{-6}$ during nighttime and to $6 \times 10^{-5} \times (J/J_{max})$ during daytime, where $J$ and $J_{max}$ [W m$^{-2}$] represent the solar intensity and its maximum value. Following the suggestion of Vogel et al. (2003) adopted, for example, by Zhang et al. (2021), we express the surface density over the ground by $1.7/h$, where $h$ [m] represents the height of the model layer adjacent to the ground; the 1.7 value represents an effective factor per ground surface area in the first layer.

In this model case, we also account for the gas-phase reactions of HONO (R24-R27) as well as the direct transportation HONO emissions. The latter are assumed to be equal to 0.8% of the traffic emission of NO (Dai et al., 2021). In this study, we neglect the direct HONO emissions from soil and the daytime HONO source from the photolysis of $NO_3^-$, which may lead to an underestimation of HONO concentration in rural areas and during daytime (Zhang et al., 2016, Fu et al., 2021; Zhang et al., 2021).

### 2.5. Photochemical reactivity and *AOC*

To characterize the oxidation capacity of the atmosphere in China, we consider several indicators that have proven to be useful for developing $O_3$-controlling strategies. These include the OH reactivity associated with the action of volatile organic compounds ($VOC^R$) and nitrogen oxides ($NO_x^R$), the radical chain length ($ChL$), the ozone production efficiency ($OPE$), and the atmospheric oxidation capacity ($AOC$).

Since $NO_x$, VOCs, and CO are oxidized by the OH radical as part of a cyclic chain process that initiates the $O_3$ formation, an estimate of the OH reactivity (expressed in s$^{-1}$) allows us to understand the factors that determine the photochemical budget of $O_3$ and more generally the factors that characterize the atmosphere's oxidizing capacity. The OH reactivity by the different VOCs and CO and by $NO_x$ is defined as



$VOC^R = \Sigma_i\, k_{10,i}\, [VOC_i] + k_{12}\, [CO],$
    (8a)

    $NO_x{}^R = k_{23}\, [NO_2],$
    (8b)


The radical chain length *ChL* provides a measure of the number of cycles affecting $RO_x$ radicals
before these radicals undergo a termination process. It can therefore be expressed by the ratio
between the conversion rate between $RO_x$ radicals, including the conversion by NO of $HO_2$ to OH
(reaction R20) and of $RO_2$ to $HO_2$ (reaction R21), and the destruction rate of $RO_x$ (or equivalently
by the production rate of $RO_x$). Thus, adopting here the definition of Martinez et al. (2003), Mao
et al. (2010), and Zhu et al. (2020), we write

$$ChL = \frac{k_{20}[HO_2][NO] + \Sigma_i k_{21,i}[RO_{2,i}][NO]}{D(RO_x)},$$
    (9)


From this adopted definition Eq. (9), and assuming that the $RO_x$ production and destruction rates
are in balance, we can write

$$P(O_x) \cong P(RO_x)ChL,$$
380    (10)

which shows that $O_x$ production is proportional to the $RO_x$ production rate and is favored by a
large number of radical regenerations.

The Ozone Production Efficiency (*OPE*) is used to quantify the efficiency of $O_3$ molecules formed
per $NO_x$ molecule oxidized. It is defined as the ratio between the $O_3$ production rate and $NO_x$ loss
rate

$$OPE = \frac{P(O_x)}{D(NO_x)} \cong \frac{P(O_x)}{P(HNO_3)},$$
390    (11)

and represents the efficiency of $NO_x$. To a good approximation, this expression can be expressed
as

$$OPE \approx \frac{k_{20}[HO_2][NO]}{k_{23}[OH][NO_2]},$$
    (12)



As the instantaneous value of *OPE* depends on the HO$_2$/OH and NO/NO$_2$ concentration ratios, it accounts for the couplings between RO$_x$ and NO$_x$ cycles. One can show that this factor is usually
highest in the remote atmosphere or low-NO$_x$ environments (Ridley, 1999). Note that, under VOC-limited conditions (polluted areas) when the production rate of odd oxygen can be expressed as (Kleinman et al., 2002)

$$P(O_x) = \sum_i \varphi_i \, [VOC_i][OH],$$
405 (13)

where $\varphi_i$ represents the O$_3$ yield from the production of HO$_2$ radicals, *OPE* can be approximated by

$$OPE \approx \overline{\varphi} \frac{VOC^R}{NO_x^R},$$
(14)

where $\overline{\varphi}$ represents an average yield value. In other words, under VOC-limited situations, the ratio between $VOC^R$ and $NO_x^R$ has some similarities with the odd oxygen production efficiency.

Finally, the atmospheric oxidizing capacity (*AOC*; expressed in cm$^{-3}$ s$^{-1}$), a parameter introduced by Geyer et al. (2001) to account for the contribution of all oxidants, is derived here as the rate at which CO, CH$_4$, and NMHCs (all species are noted here as $Y_i$) are oxidized by OH, O$_3$, and NO$_3$ (noted as $X_j$) (Geyer et al., 2001; Elshorbany et al., 2009; Xue et al., 2016; Wang et al., 2022; Yang
et al., 2022). Thus, when considering all combinations of the different primary pollutants and atmospheric oxidants, we write

$$AOC = \sum_i^j k_{i,j} \, [Y_i][X_j],$$
(15)

As stated by Wang et al. (2022) and Yang et al. (2022), *AOC* is a parameter well-suited to describe the removal rate of primary pollutants and the formation of secondary species including O$_3$ and secondary PM$_{2.5}$. It is, therefore, an indicator used to design control policies for these secondary species. During daytime, the largest contribution to *AOC* is due to the oxidation of pollutants by
the OH radical (Li et al., 2017; Liu et a., 2022). At night, the oxidizing capacity is due to the oxidation by NO$_3$ and O$_3$.

### 3. **Model description and validation**

3.1. Modeling setting



To characterize the chemical budget of reactive species, photochemical parameters and *AOC* in China, we use the version 4.1.2 of the WRF-Chem model (Skamarock et al., 2019) to simulate the meteorological fields as well as the regional transport, the chemical and physical transformations
of trace gasses and aerosols. We adopt the MOZART-4 gas-phase chemical mechanism documented and evaluated by Emmons et al. (2010), which includes 108 chemical species and 235 gas-phase reactions. This scheme is coupled to the MOSAIC aerosol module described by Fast et al. (2006), Zaveri et al. (2008), and Lu et al. (2021). A list of detailed $RO_2$, VOCs and aerosol species included in the model is provided in Table S1 in the Supplementary information.


We select one month in the winter (1st to 31st January) and the summer (1st to 31st July) of 2018 respectively to analyze the calculated distributions of chemical species. The horizontal resolution adopted in the present study is 36 km × 36 km over the entire domain that covers East and Southeast Asia (from 15° S to 60° N in latitude and 60° E to 150° E in longitude). Initial meteorological
conditions are taken from the NCEP reanalysis dataset FNL (http://rda.ucar.edu/datasets/ds083.2/). Initially chemical boundary conditions are constrained by the results of the global CAM-chem model (https://www.acom.ucar.edu/cam-chem/). The different modules used to represent physical processes are provided in Table S2.

For the anthropogenic emissions of air pollutants, we adopt the surface emissions provided by the Multi-resolution Emission Inventory for China (MEIC v1.3; http://www.meicmodel.org/) derived for the year 2017 (Zhang et a., 2009; Zheng et al., 2018). This inventory covers the anthropogenic emissions for the geographical area of mainland China. For the remaining areas of Asia, we use the anthropogenic emissions provided by the 2018 global inventory of the Copernicus Atmosphere
Monitoring Service (CAMS)-GLOB-ANT_v4.2 (Elguindi et al., 2020; Granier et al., 2019). Biogenic emissions are calculated online by the Model of Emission of Gas and Aerosols from Nature (MEGAN) version 2.1 (Guenther et al., 2006). The dust and sea-salt emissions are calculated online by the Global Ozone Chemistry Aerosol Radiation and Transport (GOCART) module (Chin et al., 2002).


The availability of several observational datasets allows us to evaluate the meteorological parameters and air pollutant concentrations derived by our regional model. The meteorological data used to validate the model simulations, including the wind direction, wind speed, surface temperatures, and specific humidity, are obtained from the NOAA National Climatic Data Center (NCDC). Con-
ventional air pollutant data, including $SO_2$, $NO_2$, CO, $O_3$, and $PM_{2.5}$, are obtained from the surface stations of China's Ministry of Ecology and Environment (MEE; https://www.mee.gov.cn/). To validate the model results, we calculate the mean bias, the normalized mean bias, the normalized mean error, the root mean square errors, and the correlation coefficient. The equations for these statistical parameters are found in the paper by Dai et al. (2020).




In our analysis presented in the subsequent sections, we examine in more detail the calculated concentrations of photochemical parameters at urban sites in four big cities in China: Beijing, Shanghai, Guangzhou, and Chengdu (Fig. 1). We also provide these parameters at four rural observational sites for which detailed observational analysis is available. These include the relatively polluted site of Wangdu in the suburban region near Beijing, the Atmospheric Supersite of Heshan located 50 km to the southwest of Guangzhou, the remote free site of Waliguan at about 3800 m altitude, and the coastal site of Hok Tsui in Hong Kong. Detailed information on the selected sites in the present study is listed in Table 2.

### 3.2. Design of numerical experiments

Table 3 lists the different sensitivity cases designed for this study. The baseline case, called *Het-all*, accounts for all heterogeneous reactions referred to in Table 1 and includes all identified sources of HONO mentioned in Sect. 2.4. The *Het-all* case is used to evaluate the performance of the model relative to observations. To quantify the specific aerosol effects through the uptake of $HO_2$, $N_2O_5$, $NO_3$ and $NO_2$ and extinction, and $NO_2$ uptake over the ground on surface $O_3$ concentration, other nine sensitivity cases are considered based on different assumptions. The details on these sensitivity cases are given as follows.

For the specific effects of aerosol uptake, the respective importance of these processes is determined by subtracting the baseline results from the results in sensitivity cases in which specific heterogeneous reactions are ignored: $HO_2$ (*No-HetHO2-Aero*), $NO_3$ (*No-HetNO3-Aero*), $N_2O_5$ (*No-HetN2O5-Aero*), $NO_2$ (*No-HetNO2-Aero*). The case labeled *No-Het-Aero* ignores all the above heterogeneous reactions on aerosols. The difference between *Het-All* and *No-Het-Aero* represents the combined effects of these heterogeneous reactions on particles. The case denoted as *No-Phot* ignores the radiative effects of aerosols on the calculation of the photodissociation coefficients. The difference between *Het-All* and *No-Phot* represents the effect of aerosol radiation. An additional case labeled as *No-Het-Aero-Phot* ignores the above-mentioned heterogeneous reactions on aerosols and aerosol effects on light extinction and photodissociation to quantify the combined effect of aerosol uptake and radiation.

In order to quantify the contribution of HONO sources added to the model, we consider a case labeled *No-HONO* in which the heterogeneous uptake of $NO_2$ by aerosols, bare soils, urban surfaces as well as the homogeneous formation, and surface emissions of HONO are all ignored. The difference between the results of the *Het-All* case and the *No-HONO* case represents the effect of all HONO sources. The final case, denoted as *No-Het-HONO-Phot,* ignores all the above heterogeneous reactions on aerosols, other HONO sources, and the radiative effects of aerosols on the calculation of the photodissociation coefficients. The difference between *Het-All* and *No-Het-HONO-Phot* provides quantitative measures of the effects resulting from all heterogeneous reactions (aerosols and ground effects), other HONO sources, and aerosol radiation.





### 3.3. Model validation

In Fig. S1, we show the spatial validation of the calculated surface concentrations of the Maximum Daily 8-hour average (MDA8) $O_3$, as well as the monthly averages $NO_2$, CO, and $PM_{2.5}$ (*Het-all* case) with available observational data from MEE for January and July 2018. In most cases, this comparison shows a good performance of the model with, however, some discrepancies: an over-estimation of summertime $O_3$ in central and western China associated with an underestimation of $NO_2$ in these regions, an underestimation of summertime $O_3$ in eastern China with a slight overes-

timation of $NO_2$ (Fig. S1). In the case of CO and $PM_{2.5}$, the calculated concentrations are higher than the measured values in central China in both seasons. The diurnal variation of $NO_2$, $O_3$, CO, and $PM_{2.5}$ in January and July for the four metropolitan areas selected in our study are compared with measurements from monitoring stations in Fig. S2 and S3. The model successfully simulates the diurnal variations of these chemicals. However, an overestimation of summertime $NO_2$ is pre-

sented in these urban areas, with an underestimation of $O_3$ at night and an overestimation during daytime. These discrepancies can be explained by the relatively lower $NO_2$ uptake coefficients used in our studies (Liu et a., 2019; Fu et al., 2019). The simulated CO is slightly overestimated, which can be attributed to uncertainties in chemical boundary conditions and emissions (Liu and Wang et al., 2020). An overestimation of $PM_{2.5}$ is found in summer, which can be partially due to

uncertainties in emissions and the mechanisms of secondary aerosol formation (Li et al., 2022).

Base estimates of the NO, HONO, HCHO, OH, $HO_2$, $NO_3$, isoprene, ethane, and ethene mixing ratios are found in Fig. S4-S6. Calculated diurnal variations of surface NO, HONO, OH, $HO_2$, and $NO_3$ are provided in Fig. S8-S11. Generally, our simulated concentrations of OH, $HO_2$, HONO, and HCHO matched relatively well with the observational data. The calculated aerosol surface

area density is shown in Fig. S12. The values calculated in eastern China are considerably higher during wintertime (2.5 to $3 \times 10^{-5}$ cm$^2$ cm$^{-3}$) than during the summer (0.7 to $1.0 \times 10^{-5}$ cm$^2$ cm$^{-3}$). A more detailed discussion and validation of the base model results are provided in the Supplementary Information.


### 4. **The budget of oxidants**

In order to highlight the regional differences in the existing photochemical regimes, we first show the distributions of areas where the $O_3$ formation is either $NO_x$- or VOC-limited. Two indicators

define these areas: the concentration between $H_2O_2$ and $HNO_3$ and between HCHO and $NO_2$. An area is considered as $NO_x$-limited if $[H_2O_2]/[HNO_3] > 0.2$ (Zhang et al., 2009) or $[HCHO]/[NO_2] > 1.0$ (Jing et al., 2021). It is assumed to be VOC-limited if $[H_2O_2]/[HNO_3] < 0.06$ (Zhang et al., 2009) or $[HCHO]/[NO_2] < 0.55$ (Jing et al., 2021). Regions in between these bounds correspond to an intermediate situation. Abdi-Oskouei et al. (2022) also define $NO_x$-sensitive region by the

criterion $L_H/L_N < 0.35$.





According to the definition based on the HCHO to $NO_2$ concentration ratios, we see in Fig. 2 that, during summertime, the ozone formation in southern and western China as well as in the surrounding oceanic regions, is primarily $NO_x$-sensitive. Exceptions are found in the south-eastern coast
with the metropolitan regions of Guangzhou and Hong Kong. The formation of ozone in a large fraction of northern China, specifically in the areas to the south of Beijing and in the region near Shanghai, is VOC-sensitive. A broader area surrounding Beijing and Shanghai is in an intermediate situation. This latter condition prevails in urban hotspots such as Seoul and Tokyo. During winter, the ozone formation in most of eastern and northern China is VOC-limited. We note that
the geographical area of the VOC-limited regime is larger when adopting the $[HCHO]/[NO_2]$ criteria than when adopting the $L_H/L_N$ ratio, but smaller than when adopting the $[H_2O_2]/[HNO_3]$ concentration ratio (see Fig. S13).

### 4.1. The budget of $RO_x$


Figure 3 shows the geographical distribution of the average daytime (08:00-19:00 Local Standard Time) production rate of $RO_x$ ($P(RO_x)$) contributed by the photolysis of HONO, HCHO, non-HCHO OVOCs and $O_3$ for the January and July of 2018.

In winter, the mean daytime production rate of the radicals is small in less populated western China. In the eastern plain, its value associated with the HONO photolysis is typically 0.5-0.8 ppbv $h^{-1}$ in rural areas and reaches 1-2 ppbv $h^{-1}$ in polluted urban areas. The contribution of the HCHO photolysis to $P(RO_x)$ is of the order of 0.15 ppbv $h^{-1}$ in most areas, with values as high as 0.5 ppbv $h^{-1}$ in Guangzhou, which benefits from sufficient solar radiation during winter. Correspondingly,
the contribution of other OVOCs is around 0.2 ppbv $h^{-1}$ in southern China, with a similar distribution of the photolysis of HCHO. The mean daytime $P(RO_x)$ associated with $O_3$ photolysis is small (less than 0.5 ppbv $h^{-1}$) over the entire Chinese territory except in the very south of the country, where it reaches 0.2 ppbv $h^{-1}$. The contribution by alkene ozonolysis is negligible (Fig. S14).

In summer, the mean $P(RO_x)$ by HONO photolysis reaches 2 to 3 ppbv $h^{-1}$ in the regions surrounding Beijing, Shanghai, Guangzhou, and Chengdu but is considerably smaller (less than 0.5 ppbv $h^{-1}$) in the outskirt rural areas. The photolysis of HCHO reaches values ranging from 0.5 to 1.0 ppbv $h^{-1}$ in the rural areas of eastern China, with no particular maximum in metropolitan areas. The high value of $P(RO_x)$ contributed by non-HCHO OVOCs photolysis, ranging from 0.3 to 0.8
ppbv $h^{-1}$, with obvious peak values in city areas. The daytime averaged value of $P(RO_x)$ resulting from the photolysis of $O_3$ is of the order of 1.0 ppbv $h^{-1}$ in eastern and southern China. The peak spot in the Sichuan basin is due to the high water vapor contributed by heavy rainfall in summer, which leads to high OH radical (Xia et al., 2021).



In summary, and for the present conditions, our model suggests a higher value of $P(RO_x)$ in sum-
       mer than in winter. The higher summertime $P(RO_x)$ in eastern and southern China is associated
       with the photolysis of formaldehyde and $O_3$. In urban areas, the dominant contribution to the higher
       $P(RO_x)$ in summer is provided by the photolysis of HONO. The spatial distributions that vary in
       winter and summer are also related to the seasonal variations in meteorological parameters, such
as surface temperature and water vapor (Fig. S15).

       The diurnal variation of the $P(RO_x)$ in four different metropolitan areas (Beijing, Shanghai, Guang-
       zhou, and Chengdu) at two relatively polluted rural (Wangdu and Heshan) and two clean remote
       sites (Hok Tsui and Waliguan) is shown for summertime in Fig. 4. The graph shows the contribu-
tion of the HONO (green area), HCHO (red area), non-HCHO OVOCs (dark green area), and $O_3$
       (yellow green area) photolysis, as well as the effect of alkene ozonolysis (blue area). In the four
       urban centers, the maximum values of $P(RO_x)$ in the early afternoon range between 5 ppbv h$^{-1}$ in
       Shanghai and 6-8 ppbv h$^{-1}$ in the three other cities. In the early morning, as the sun rises, the largest
       contribution is due to the photolysis of HONO. A few hours later, the contribution of the photolysis
of HCHO and other OVOCs becomes large. The value of $P(RO_x)$ from the photolysis of $O_3$ is
       small in the early morning and peaks in the mid-afternoon. In the four sites of the rural areas, the
       maximum value of $P(RO_x)$ is close to 5 ppbv h$^{-1}$ in Wangdu (suburban site southwest of Beijing),
       1.3 ppbv h$^{-1}$ in Heshan (rural site close to Guangzhou), and less than 0.3 ppbv h$^{-1}$ in Hok Tsui
       (coastal site) and Waliguan (western China). The contribution of $O_3$ photolysis generally domi-
nates at these rural sites.

       A comparison between the values of $P(RO_x)$ derived from local observations is performed with
       model estimates in Table 4. At four city sites, our simulated values of the maximum of $P(RO_x)$
       and the contribution by the photolysis of HONO, HCHO and $O_3$ reproduce satisfactorily the ob-
servations. However, at the Heshan site, the calculated value of $P(RO_x)$ (1.1 ppbv h$^{-1}$) in our study
       is much smaller than the observed value (4.0 ppbv h$^{-1}$) (Tan et al., 2019). One reason for this
       discrepancy may be the missing soil HONO emission, which leads to an underestimation of HONO
       concentration at this site (Table S4) and the contributed value to $P(RO_x)$ by the photolysis of
       HONO (0.5 ppbv h$^{-1}$ v.s. 2.0 ppbv h$^{-1}$). Another uncertainty is the contribution of non-HCHO
OVOCs photolysis to the $P(RO_x)$. Wang W. et al. (2022), based on the measurement data in
       Guangzhou, reported that a model without constraints of non-HCHO OVOCs would lead to an
       underestimation in the production rate of $RO_x$ and $O_3$. Owing to the lack of specific OVOCs meas-
       urements, it is difficult to quantify the contributions of different OVOC species. Additional efforts
       regarding the OVOC measurements are needed to understand the specific contribution of OVOCs
to the atmospheric oxidation capacity.

       The diurnal variations of $P(RO_x)$ are displayed in Fig. S16. The maximum value of the $P(RO_x)$ is
       close to 3 ppbv h$^{-1}$ in Beijing, Chengdu and Shanghai and is about twice as large in the region of
       Guangzhou. In Wangdu and Heshan, the peak values are comparable to those in urban areas, while





in Hok Tsui and Waligan, they are lower than 1 and 0.1 ppbv h$^{-1}$, respectively. In most cases, the major contribution to $P(RO_x)$ is provided by the photolysis of HONO.

The spatial distribution of the daytime averaged destruction rate for $RO_x$ ($D(RO_x)$) in January and July is shown in Fig. 5. In the two seasons under consideration, the loss rate is the largest in the

eastern and southern regions of China. In January, this quantity is largest in polluted metropolitan areas, with daytime mean values surpassing 2 ppbv h$^{-1}$. In July, the total daytime average value of $D(RO_x)$ is of the order of 3 ppbv h$^{-1}$ in the rural areas of eastern China and reaches about 6 ppbv h$^{-1}$ in the urban and industrialized areas. In Tibet, the value of this quantity is small (~0.1 ppbv h$^{-1}$ in winter and 1.2 ppbv h$^{-1}$ in summer).


Interestingly, the relative importance of the different photochemical mechanisms involved in the $D(RO_x)$ varies considerably with the season. In January, the $D(RO_x)$ due to the reaction between OH radicals and $NO_x$ ($L_N$ in Eq. 2) dominates in most parts of eastern China (contribution of typically 90%) except in Tibet, where the largest loss (contribution of 70-80%) is due to the recom-

bination of hydrogen radicals ($L_H$ in Eq. 2). In July, it is this last type of loss ($L_H$) that plays the dominant role (typically 90%), except in the eastern plain of China where the level of $NO_x$ is highest, and $L_N$ (contribution of 70-80%) is, therefore, larger than $L_H$ (contribution of 20%). In both seasons, the destruction of $RO_x$ by the uptake of $HO_2$ is relatively small (generally less than 15%). The highest contribution occurs during winter in southwestern China (contribution of 30%)

and in the Ganges River Valley of India (contribution of 40%).

The diurnal variation of $D(RO_x)$ in July is presented in Fig. 6 in four urban areas (Beijing, Shanghai, Guangzhou, and Chengdu) and in four selected locations in rural areas (Wangdu, Heshan, Hoktsui, and Waliguan, see Fig. 1). In the summertime, the value of $D(RO_x)$ around noon reaches

about 13 ppbv h$^{-1}$ in the urban areas except in Shanghai, where it reaches only 8 ppbv h$^{-1}$. In winter, the corresponding maximum values are closer to 3-4 ppbv h$^{-1}$, except in the southern city of Guangzhou, where the maximum loss rate is closer to 6 ppbv h$^{-1}$. In all these cases, the dominant contribution to $D(RO_x)$ is attributed to the reactions involving the presence of $NO_x$ ($L_N$). In rural areas, the value of $D(RO_x)$ is considerably smaller. In July, it is of the order or smaller than 2 ppbv h$^{-1}$

and is dominated by the $HO_x$ recombination ($L_H$). In the wintertime, the peak loss is smaller than 0.2 ppbv h$^{-1}$ except in Chengdu, where it reaches 0.4 ppbv h$^{-1}$. The major contribution is due to the reactions involving $NO_x$ ($L_N$).

The experimental study of Whalley et al. (2021) in Beijing during the summer of 2018 provides

for $D(RO_x)$ a maximum daytime value of 7 ppbv h$^{-1}$ with the following contributions: 4 ppbv h$^{-1}$, 1.3 ppbv h$^{-1}$, and 1 ppbv h$^{-1}$ for the $NO_2 + OH$, $NO + OH$, and $RO_2 + OH$ reactions, respectively. Our simulated value matches well the reported experimental data, and the corresponding values in July are 5 ppbv h$^{-1}$, 0.8 ppbv h$^{-1}$, and 1.2 ppbv h$^{-1}$, respectively. Yang et al. (2021) report diurnal variations in $D(RO_x)$ as derived from their observation in Chengdu during the autumn of 2018.





The peak value of this quantity is about 7 ppbv, which is lower than our calculated value of 12.5 ppbv h$^{-1}$ in July. The higher level in our study is due to the overestimated concentration of summertime NO$_2$ in Chengdu (Fig. S2). This overestimation also leads to a higher contribution of the NO$_2$ + OH reaction by 55% in our study than the reported 35% in the literature.

**4.2. The budget of odd oxygen**

The production rate of odd oxygen ($P(O_x)$) with its two contributions (reaction of NO with hydrogenated and organic peroxy radicals (HO$_2$ and RO$_2$) shown in Eq. 3) is shown in Fig. 7 for January and July 2018. In the Northern China Plain and other urbanized areas, the production rate is of the

order of 4-6 ppbv h$^{-1}$ during winter (January), while in the rural areas of southern China, it is larger than 20 ppbv h$^{-1}$ during summer (July) and 6-10 ppbv h$^{-1}$ during winter. The value of $P(O_x)$ is very small in the western part of China. The relative contributions of both step-limiting processes to the total $P(O_x)$ are of the same order of magnitude, although the reaction involving the hydrogenated peroxy radicals seems to slightly dominate, particularly outside densely populated areas.


The diurnal variations in $P(O_x)$ are depicted in Fig. 8 for specific areas of China in July. This graph highlights the maximum values found during the early afternoon in urban areas: 115 pphv h$^{-1}$ in Beijing, 40 pphv h$^{-1}$ in Shanghai, 110 ppbv h$^{-1}$ in Guangzhou, and 70 ppbv h$^{-1}$ in Chengdu. In Beijing, Whalley et al. (2021) derived from their observations in the summer of 2018 a maximum

O$_x$ production rate of 100 ppbv h$^{-1}$. These high values must be contrasted by the considerably lower values found in rural areas: 1.5 ppbv h$^{-1}$ at Mount Walinguan and Hok Tsui. Intermediate maximum values are found at the sites located in the vicinity of large metropolitan areas: 40 ppbv h$^{-1}$ in Wangdu and only 7 ppbv h$^{-1}$ in Heshan. The graph also shows the relative contribution of the hydrogen and organic peroxy radicals. Both radicals contribute about equally to the odd oxygen

production rate. The contribution of the organic peroxy radical is determined by anthropogenic emissions of hydrocarbons in the cities and by biogenic hydrocarbons in rural areas. A similar representation of the factors contributing to the formation of O$_3$ during winter is shown in Fig. S17.

Finally, a quantitative estimate of Eq. (4) is provided in Fig. 9. In the eastern regions of China, the largest contribution to the diurnal mean value of $D(O_x)$ is due to the reaction between NO$_2$ and OH, particularly in winter. This chemical path remains, however, the dominant loss channel during summer in the polluted northern plain between Shanghai and Beijing. In urbanized area, the reaction between H$_2$O and O($^1$D) also play a relatively considerable role on the value of $D(O_x)$ in

summer. The relative contribution of ozonolysis reaction with alkene to the value of D(O$_x$) is displayed in the southern China, which is associated with the high level of alkene in this area. In rural area, the highest contribution is from the reaction between H$_2$O and O($^1$D) in summer, followed by the reaction between HO$_2$ and O$_3$.



## 5. **Effects of aerosols on oxidants**

The presence of aerosols in the atmosphere affects the abundance of atmospheric oxidants primarily through two different processes: (1) changes in the heterogeneous reaction rates associated with the uptake of several species by the particles (Tan et al., 2020; 2022), and (2) changes in the photolysis rate associated with enhanced extinction of solar light (Tie et al., 2001; 2005; Xing et al., 2017; Tan et al., 2022). Here we assess the relative importance of these two different mechanisms and derive the combined effect on the concentration of surface $O_3$.

### 5.1. Effects due to heterogeneous reactions

Figure S18 summarizes the response of surface NO, $NO_2$, OH, and $HO_2$ concentrations to the introduction of the added heterogeneous chemical reactions (R28-31 in Table 1) in the model. The concentration of $NO_x$ species decreases due to the enhanced conversion of $NO_x$ into $HNO_3$. In the eastern plain of China, we derive a reduction of up to 9 ppbv for $NO_2$ and 3 ppbv for NO in winter, with a summertime decrease of 6 ppbv for $NO_2$ and 2 ppbv for NO. At the same time, the concentration of $HO_x$ increases due to the enhanced formation of HONO, which is a source of $HO_x$ in the presence of sunlight. This process overrides the expected reduction in $HO_2$ due to its uptake by the aerosol. We find an increase of up to 0.15 pptv for OH and 5 pptv for $HO_2$ in winter, and 0.3 pptv for OH and 8 pptv for $HO_2$ in summer.

We now examine how the uptake of $HO_2$, $N_2O_5$, and $NO_2$ on the surface of particles modifies the surface concentration of surface $O_3$ (Fig. 10). As shown by Fig. 10a, the uptake of $HO_2$ onto aerosols in January leads to a reduction in the surface concentration of ozone of about 3-4 ppbv, with the large decrease concentrated in Sichuan Basin and central China. In July (Fig. 10b), the highest ozone changes are found in the North China Plain, especially in the vicinity of Beijing (about 3 ppbv). The varied high spots of aerosols effect of $HO_2$ uptake on ozone is associated with the distribution of aerosol surface area density. In winter (Fig. S12), the simulated high value of aerosol surface area density is derived in Sichuan Basin and central China, while, in summer, the high value is calculated in Beijing and surrounding areas. This indicates the high sensitivity of the effect of $HO_2$ uptake on particles to the aerosol geometric parameters (Song et al., 2020).

The response of ozone to the uptake of $N_2O_5$ by aerosols is negative during winter when the competing photochemical conversion of $NO_x$ to $HNO_3$ by the OH radical is very slow. The heterogeneous conversion of $N_2O_5$ to $HNO_3$ tends to reduce ozone by up to 3-4 ppbv in southern China during winter (Fig. 10c), with limited effects in the summertime (Fig. 10d).

The uptake of $NO_2$ by aerosols tends to increase the wintertime ozone concentration by 8-9 ppbv in eastern China and in large urban areas of southern China (Fig. 10e) since the photolysis of HONO (formed from the heterogeneous $NO_2$ conversion) leads to enhanced concentrations of NO



and OH. As the simulated value of wintertime $HO_2$ is low (below 0.5 pptv) in large parts of China, the production of $HO_x$, from the photolysis of HONO, dominantly controls the value of $P(RO_x)$ (Fig. S14) and the formation of $O_3$ in this season. In summer (Fig. 10f), the concentration of ozone is reduced by 3-6 ppbv in the $NO_x$-sensitive rural areas of eastern and central China but is enhanced by 6-7 ppbv in VOC-sensitive urban areas. During this season, the high value of the $HO_2$ density

weakens the contribution of the $HO_x$ produced by the HONO photolysis. However, the lower level of summertime $NO_x$ strengthens the effect of the $NO_2$ loss resulting from the $NO_2$ uptake on particles.

The effect on near-surface ozone of the heterogeneous conversion of $NO_3$ by aerosols, also con-

sidered in the present model study (not shown), has been found to be very small.

The lowest panels of Fig. 10 show the change in ozone resulting from all four heterogeneous processes on aerosol surfaces. When combining the effects of all $HO_2$, $NO_2$, $NO_3$, and $N_2O_5$ heterogeneous reactions, we derive an ozone increase of 6-8 ppbv in winter and a decrease of 6-8 ppbv

in summer. However, ozone increases up to 8 ppbv in the VOC-limited metropolitan areas of Beijing, Shanghai, Guangzhou, and Chengdu. Comparison of Fig. 10 e, f with Fig. 10 g, h, respectively, suggests that heterogeneous loss of $NO_2$ on atmospheric particles discussed above dominates the impact of the studied heterogeneous reactions on surface $O_3$.

The $O_3$ response to the uptake of $HO_2$ and $NO_2$ is complex. The $HO_2 + O_3$ reaction provides a direct destruction mechanism for ozone, and the heterogeneous uptake of $HO_2$ contributes, therefore, to ozone enhancement. At the same time, the $HO_2$ uptake reduces the ozone production resulting from the reaction between $HO_2$ and NO, a photochemical process that is most efficient during summertime. The conversion of $NO_2$ to $HNO_3$ tends to reduce the $O_3$ formation in $NO_x$-

limited areas due to the loss of $NO_2$ by particles. However, in VOC-limited areas, the loss of $NO_2$ leads to an increase in the $O_3$ concentration. Moreover, the photolysis of HONO, which results from $NO_2$ uptake, produces NO and OH, which further affects the formation of $O_3$.

Our model simulation in July suggests that the presence of aerosol leads to a decrease of $O_3$ in

$NO_x$-limited areas and an increase of $O_3$ in VOC-limited areas. In other words, the continuous reduction in aerosol emissions observed in the past years should have led to an increased ozone concentration in $NO_x$-limited areas and a reduced ozone concentration in VOC-limited areas. The $O_3$ decrease in VOC-limited areas is the result of two opposite effects: the ozone decrease due to reduced $NO_2$ uptake (increased $NO_x$ densities and enhanced ozone titration), and the ozone in-

crease from reduced $HO_2$ uptake (increased $HO_2$ concentration and enhanced rate of the $HO_2 +$ NO reaction). The importance of $HO_2$ uptake by aerosols on ozone formation has been highlighted by several modeling studies (Li et al., 2021; Liu and Wang, 2021; Ivatt et al., 2022). However, recent studies based on field measurements (Tan et al., 2020; 2022; Dyson et al., 2022; Yang et





al., 2021) made in the urban/rural areas of northern and southern China (Wangdu, Beijing, Shen-
zhen, and Chengdu) during the summertime of 2014, 2017, 2018, and 2019, highlighted the minor
importance of $HO_2$ uptake for radical chemistry and $O_3$ formation, and showed the increasing im-
portance for the ozone production of the reduced $NO_2$ uptake by particles. One potential reason
for changes in the conclusions of these studies could be attributed to the sharp reduction in the
emissions of pollutants in China (Zheng et al., 2018), including the reduction in the aerosol load
and in the anthropogenic $NO_x$ emissions. Another possible explanation is the adoption for the
analyses of different values for the uptake coefficients and for the aerosol geometric parameters
associated with the heterogeneous reactions affecting $NO_2$ and $HO_2$.

### 5.2. Effects due to photolysis

The presence of aerosols in the atmosphere tends to enhance the absorption and scattering of in-
coming solar radiation with direct impacts on the photolysis rates and hence on the abundance of
chemical species. Figure S19 shows a model estimate of the resulting effects on the surface con-
centrations of $NO_2$, $NO$, $OH$, and $HO_2$.

In the month of January, during which the aerosol burden is high, and the solar intensity is low,
the effect of light reduction by the aerosols through changes in the photolysis rates tends to increase
the surface concentration of $NO_x$, especially in the most populated and polluted urban areas (Bei-
jing, Shanghai, and Chengdu) where an increase in the concentration of $NO_2$ typically 0.5 ppbv is
derived. In these urban areas, the concentration of $NO$ is increased by 0.5 ppbv. A reduction in
surface $OH$ (about 0.1 pptv) and $HO_2$ concentrations (about 1.5 pptv) is derived, with the largest
effect occurring in the southeastern regions of China. The surface $O_3$ decreases by up to 4-5 ppbv
(Fig. 11a), with the highest decrease found in the Sichuan basin. In July, the aerosol burden is
lower, while the solar intensity is higher. The effect of light reduction by the aerosols tends to
increase the surface concentration of $NO_2$ by 1 ppbv in the North China Plain. In the case of $OH$
and $HO_2$, a decrease of 0.05-0.1 pptv and 2-3 pptv is found in the North China Plain. A decrease
of $O_3$ by up to 3-4 ppbv is derived in the Beijing and surrounding area (Fig. 11b).

### 5.3. Combined aerosol effects on ozone (uptake effects and photolysis).

When all heterogeneous processes affecting $HO_2$, $N_2O_5$, and $NO_2$ are simultaneously taken into
account, we derive for January an increase in the surface ozone concentration that is generally of
the order of 6-8 ppbv (Fig. 11c) in the middle of the country. The change in the photolysis rates
reduces the ozone concentration by 2-4 ppbv (Fig. 11a) and compensates to some extent the in-
crease due to aerosol uptake. Such a compensation mechanism was highlighted by Tan et al. (2022)
based on observations made in Shenzhen in the autumn of 2018. The combined effect in winter is
therefore limited, with ozone values increasing by less than 4 ppbv in most regions of China and
by less than 6 ppbv in the urban area.



In July, when combining the photolysis and uptake effects (Fig. S20), we derive a decrease in the concentration of $NO_2$ (up to 10 pptv) and NO (up to 5 pptv) and an increase in the concentration of OH (0.05 pptv) and $HO_2$ (up to 10 pptv) in eastern China. The response of ozone (Fig. 11b and d) is characterized by a reduction in the surface concentration of about 10 ppbv (or 15%) in most $NO_x$-limited regions of China. In the metropolitan areas of Shanghai and Beijing, an increase of

about 8 ppbv (or about 12%) is calculated.

    These results highlight that, during summertime, the presence of aerosol particles leads to a decrease of the surface $O_3$ concentrations in $NO_x$-limited areas whereas it produces an increase in the ozone level in the VOC-limited (metropolitan) areas. Figure 12 presents a schematic view of dif-

ferent pathways that characterize the effects of aerosols on ozone concentrations. Specifically, this figure suggests that the reduction in the aerosol burden that has occurred in China in recent years should have produced an increase in surface ozone concentrations in $NO_x$-limited areas and a decrease in VOC-limited areas. The cause of the ozone increase in $NO_x$-sensitive areas should not be attributed exclusively to a reduction of the $HO_2$ uptake but to a combination of the different

uptake processes and a reduction in the light extinction by the aerosols. The ozone decrease in VOC-limited areas should be mainly attributed to the $NO_2$ uptake with a counteracting effect by the $HO_2$ aerosol uptake and by the light extinction by the particles. Our results imply that, if the aerosol loading continues to decrease in the future, the ozone formation will increase so that the air quality measures currently implemented will become less efficient in $NO_x$-limited areas. This

does not imply that ozone will necessarily increase in VOC-limited areas. With a further reduction in the $NO_x$ emissions, which tends to shift the $O_3$ formation regimes from VOC-limited to $NO_x$-limited (Tan et al., 2022), the $O_3$ response to aerosol effects may gradually reverse in these geographical areas.

5.4. Effects of other HONO sources on ozone.

    Figure 11e-f shows the changes in the surface ozone concentration due to all sources of HONO considered in the model, including direct emissions from transportation, gas phase production, and heterogeneous reactions of $NO_2$ uptake on aerosol surfaces and on ground surfaces. In January,

the increase of the ozone concentration due to all these different processes reaches 9 ppbv and is more pronounced than in the "aerosol-only" case shown in Fig. 10e. In July, the decrease in the ozone concentration in eastern China (8-10 ppbv) and the increase in the metropolitan regions of Beijing and Shanghai (6-8 ppbv) are about 50% larger than when only the aerosol uptake of $NO_2$ is taken into account (see Fig. 10f).


    Figure 11g-h also shows the changes in the surface ozone concentrations when all heterogeneous reactions involving $HO_2$, $NO_2$, $NO_3$, and $N_2O_5$ as well as all sources of HONO are included in the model calculation. These two panels (g and h) must be compared with panels (c and d) of Fig. 11.





With the additional formation processes of HONO, surface ozone is increased by about 6-8 ppbv
in southeastern China during winter. For summer conditions, surface ozone concentration is re-
duced by up to 10 ppbv in the eastern and southern parts of China, but is increased by about 6-8
ppbv in the two major metropolitan centers.

6. **Quantification of the oxidizing capacity of the atmosphere in China**

6.1. OH reactivity

The model results presented above allow us to quantify the different factors that characterize the
oxidizing capacity of the atmosphere in China. We first analyze the geographical distribution of
the OH reactivity (Eq. 8a and 8b) resulting from the reaction of this radical with VOCs and CO
(noted $VOC^R$ (Fig. 13a-b)) as well as NO$_x$ (noted $NO_x^R$ (Fig. 13c-d)). These quantities, and particu-
larly the $VOC^R/NO_x^R$ ratio (Fig. 13e-f), can be viewed as a proxy representing the competition
between radical production and destruction (see Kirchner et al., 2001).

During winter, the calculated value of the daytime averaged $VOC^R$ ranges from typically 2 s$^{-1}$,
mostly in rural areas, to 10 s$^{-1}$ in the North China Plain between the urban areas of Shanghai and
Beijing, as well as in the area of Chengdu. The high value of calculated $VOC^R$ in urbanized areas
is consistent with high values in the spatial distribution of wintertime VOCs, such as ethene (Fig.
S5), ethane (Fig. S5), and HCHO (Fig. S4). The values derived for the daytime averaged $NO_x^R$ are
of the order of 10 s$^{-1}$ in the North China Plain and most metropolitan areas of the country. Values
are close to 1 s$^{-1}$ in rural areas. The $VOC^R/NO_x^R$ ratio is of the order of 2 in most regions of China
except in the polluted areas where values close to 0.6 to 1 are derived.

During summer, the daytime averaged $VOC^R$ parameter reaches values close to or higher than 10
s$^{-1}$ in southern China. This distribution of these high values is consistent with the spatial distribu-
tion of isoprene (Fig. S5) and HCHO (Fig. S4). The value of $NO_x^R$ is smaller than in wintertime
with values generally close to 5 s$^{-1}$ in the North China Plain, and approaching 10 s$^{-1}$ inside the
cities of Beijing, Shanghai, Guangzhou, and Chengdu. The $VOC^R/NO_x^R$ ratio is larger than 2 in the
entire spatial domain, except in a small polluted area of the North China Plain and in the urban
areas of Guangzhou and Shanghai where it is close to 1. Figure S21 shows the diurnal variation of
the simulated value of the $VOC^R$ to $NO_x^R$ ratio at different sites in January and July of 2018. The
daytime values range from 0.5 to 1.8 in urban sites in both two months, with the highest daytime
value shown in the Wangdu site in July (by the value of 5).

The diurnal variations of the OH reactivities due to different organic compounds, CO and nitrogen
dioxide in January and July are shown in Fig. 14. These calculated values need to be compared
with the data provided by the observations. In city sites, for example, Whalley et al. (2021) reported
diurnal variations of the OH reactivity in Beijing during the summer of 2018 with values of 25-35



s$^{-1}$ to be compared to our model values of 23-42 s$^{-1}$ in July. Based on measurements, the contribution to the OH reactivity of NO$_x$ reactions is 40-50%, and of VOCs reactions 40-50%. The corresponding values derived in our model study in July are 50% and 45%, respectively. In Shanghai during summertime, Zhu et al. (2021) derived values of 10-25 s$^{-1}$, where the contribution of the reaction of OH with NO$_x$ is 33%, with CO is 26%, with OVOCs is 18%, and with alkenes is 15%. The reactivity value derived by our model is 10-17 s$^{-1}$, with the highest contribution from the reaction of OH with NO$_x$ (50-60%). The values measured by Tan et al. (2018) in Chengdu (September 2016) are in the range of 15-30 s$^{-1}$ to be compared to our July model values of 12-32 s$^{-1}$. During the campaign that took place in Shenzhen in the autumn of 2018, Yang et al. (2022) derived a total OH reactivity value that varied between 10 and 25 s$^{-1}$, which is relatively lower than our calculated value by 20-45 s$^{-1}$ in Guangzhou in summer.

Several other attempts have been made to derive the OH reactivity from in situ observations in rural sites. At Wangdu during summertime, Tan et al. (2017) derived a reactivity value of 12-23 s$^{-1}$, and our model study provides a value of 8-22 s$^{-1}$. Tan et al. (2019), refer to the campaign conducted at the Heshan site in the autumn of 2014, derived experimentally mean daytime OH reactivities that range from 20 to 40 s$^{-1}$, with the contributions with CO, NMHCs, and NO$_x$, are 10% (2-4 s$^{-1}$), 20% (4-8 s$^{-1}$), and 14% (3-6 s$^{-1}$), respectively. The value of $VOC^R$ and $NO_x^R$ in our model account for approximately 1 to 5 s$^{-1}$ and 2 to 5 s$^{-1}$, respectively. The underestimation of the HONO concentration in our model (Table S5), contributes to the underestimation of the calculated value of the OH radical and the OH reactivity.

Generally, in winter (Fig. 14a), the dominant contribution to OH reactivity in urban/rural sites is through the reaction of OH and NO$_x$ (40~60%), while, at a remote site (Waliguan) is from the reaction between OH and CO (80%). In summer (Fig. 14b), the contribution of NO$_x$ to OH reactivity is still high in urban/rural sites (40~70%), with one exception at the Heshan site. At this site, the largest reactivity of OH results from the reaction with alkene (40~50%), which is associated with the relatively low value of NO$_x$ (Fig. S1d) and high value of isoprene (Fig. S5) at this site.

We also show in Fig. 15 three other indicators; these describe the catalytic cycling of NO$_x$ leading to ozone formation until the chain reaction is interrupted: the radical chain length $ChL$ (defined by Eq. 9), the ozone production efficiency $OPE$ (defined by Eq. 11), and the atmospheric oxidation capacity $AOC$ (defined by Eq. 15).

### 6.2. *ChL*

The value of the daytime averaged $ChL$, which, according to our definition, increases with the atmospheric concentration of RO$_x$ radicals and NO, is of the order of 3 to 5 cycles in remote areas. In January (Fig. 15a), $ChL$ reaches values as high as 8 to 10 cycles in the southern area of China. The $ChL$ values are low in northern and western China, where the NO concentrations are low



during winter. In July (Fig. 15b), the highest values of *ChL* (about 10 cycles) are found only in the
metropolitan areas, where the $HO_x$ concentrations are high. Since this parameter can be viewed as
the ratio between the ozone and $RO_x$ production rate (see Eq. 11), the polluted areas tend to favor
ozone production for a given value of the $RO_x$ formation rate. Examples of the diurnal variation
of the chain length are provided in Fig. S22 in the Supplementary Information.

A few experimental estimates are available to quantify the value of the chain length: Zhu et al.
(2021) derived a value of 2-6 for daytime *ChL* in Shanghai during the warm season of 2018 to be
compared to the values of 4-5 provided by our model (Fig. S22). Yang et al. (2021) derive daytime
values of 2-4.5 in Chengdu during summer, while our model provides values close to 4-5. In
Guangzhou during summer, Wang et al. (2022) derive daytime values of 3-12 in fair agreement
with our calculated values of 6-10.

### 6.3. Ozone production efficiency (*OPE*)

The Ozone Production Efficiency (*OPE*) represents the number of ozone molecules produced by
$NO_x$ before $NO_x$ is further oxidized to form more stable nitrogen reservoirs or removed from the
atmosphere by deposition. The daytime averaged *OPE* values are highest (larger than 30) in the
southwest of China and are of the order of 25-30 above the Tibetan plateau (Fig. 15C-d) in both
seasons. *OPE* values typically ranges between 3 to 15 in the eastern plain during both months
under consideration. We note the similarities suggested by Eq. (14) between the distributions of
*OPE* and of the $VOC^R/NO_x^R$ ratio in the VOC-limited regions of the China Northern Plain and over
the Eastern China Sea where the effects of ship emissions are visible. Daytime *OPE* values are
usually low at urban sites. More details on the diurnal variation of the *OPE* are provided in Fig.
S23 of the Supplementary Information.

A few experimental data characterizing the *OPE* are available: Wang et al. (2017) summarize the
summertime *OPE* values derived in China between 2006 and 2015 with values ranging from 2.1
to 20.2, which is comparable with our results that range from 1 to 30.

### 6.4. Atmospheric Oxidizing Capacity (*AOC*)
Finally, we show in Fig. 15e-f the spatial distribution of the daytime averaged value of the atmos-
pheric oxidizing capacity *AOC* in January and July, respectively. In winter, the highest daytime
values of *AOC* are found in the southern part of China, especially in the Pearl River Delta region
($3\times10^7$ cm$^{-3}$ s$^{-1}$). During nighttime (Fig. S24), the *AOC* values are lower than $0.2\times10^7$ cm$^{-3}$ s$^{-1}$,
with maximum values found at the southern coast of China. These high *AOC* values are associated
with the spatial distribution of wintertime formaldehyde (Fig. S4), isoprene (Fig. S5), and of the
$HO_x$ and $NO_3$ radicals (Fig. S6). In summer, the values of daytime *AOC* are highest in the metro-
politan urban areas (up to $10\times10^7$ cm$^{-3}$ s$^{-1}$), particularly in the vicinity of Beijing. The nocturnal



*AOC* values are lower than $2\times10^7$ cm$^{-3}$ s$^{-1}$ (Fig. S24) with high spots are found in urban areas..
The distribution of summertime *AOC* has some resemblance with the distribution of nitrogen species, including NO$_2$ (Fig. S1) and HONO (Fig. S4).

Figure 16 shows the diurnal evolution of the *AOC* and the dominant photochemical processes that contribute to this quantity in different urban and rural areas for January and July, respectively.
First, we note that, in January, the noontime value of *AOC* does not supersede $6\times10^7$ cm$^{-3}$ s$^{-1}$ in urban areas and $3\times10^7$ cm$^{-3}$ s$^{-1}$ at rural sites (where the value is lower than $1.5\times10^7$ cm$^{-3}$ s$^{-1}$ in most cases). At the remote high-altitude station of Waliguan in western China (remote conditions), the maximum *AOC* value is lower than $0.2\times10^7$ cm$^{-3}$ s$^{-1}$. In July, as expected, the oxidizing capacity is larger than in winter, with noontime values reaching $(15\text{-}20)\times10^7$ cm$^{-3}$ s$^{-1}$ in metropolitan areas
but limited increase in rural and remote areas. The increasing summertime *AOC* in city sites is attributed to the larger value of *AOC* contributed by OVOCs and alkene, which is associated with the higher value of summertime OVOCs (Fig. S4) and isoprene (Fig. S6) in urban areas.

Additional information on the relative contribution of different photochemical processes as a function
of the time of the day and for two seasons is provided in Fig. 17. During winter, in Beijing during daytime, the major contributions to *AOC* are provided by the reaction of OH with alkenes, aromatics, and carbon monoxide. The situation is similar in the other three metropolitan areas under consideration. In Chengdu, however, the relative contribution of CO is larger as is the case at rural sites where the concentration of hydrocarbons is generally low. At night, the largest con-
tribution in Beijing, Shanghai, and Chengdu is provided by the oxidation of hydrocarbons by ozone.

During summertime, the daytime value of *AOC* in urban regions is determined by the reaction of OH with alkenes and with OVOCs. In remote areas, the dominant contribution is attributed to the
reaction of OH with OVOCs and with CO. OVOCs are produced as a result of the oxidation of biogenic hydrocarbons such as isoprene, primarily but not exclusively in rural areas, and by the oxidation of anthropogenic hydrocarbons, mostly in urban and industrialized areas. Thus, any analysis of the processes that determine the value of *AOC*, particularly during summertime, must take into account the role played by OVOCs. Li et al. (2023) reported that the OVOCs have a significant
impact on the atmospheric oxidative capacity in the Yangtze River Delta region, through reactions with OH during daytime and with NO$_3$ at night. Due to the limited amount of measurement data available (Wang W. et al., 2022), and the high uncertainties in the emissions of organic species (Li et al., 2023), the contributions of specific OVOCs to the oxidative process are still not unclear. More work is needed to obtain a better understanding of the impact of OVOCs on the oxidative
processes in China.

In all regions under consideration, during summertime, a dominant nighttime contribution to the value of *AOC* is provided by the oxidation of hydrocarbons by NO$_3$. As the production of NO$_3$





occurs through the reaction of $O_3$ and $NO_2$, the higher summertime $O_3$ concentrations and temper-
ature lead to the larger formation of $NO_3$ than during winter. Wang H. (2023) highlights the in-
creasingly critical role of $NO_3$-related nighttime oxidative chemistry associated with the positive
trend in particulate nitrate abundance and in the formation of $O_3$ and other secondary pollutants.
With the reduction in $NO_x$ emissions, the importance of $NO_3$ radicals may become more notable.

The model study confirms that, during daytime, more than 90% of *AOC* is due to reactions of
chemical species with the OH radical. During nighttime, the oxidation processes are considerably
slower and are due principally to the reactions of hydrocarbons with the nitrate radical ($NO_3$) and
alkenes with ozone ($O_3$). In urban areas, the dominant daytime contributions to *AOC* during sum-
mertime are the reactions with carbon monoxide (20%), alkenes (35%), aromatics (10%), and
OVOCs (30%). During winter, the corresponding numbers are 25% for carbon monoxide, 40% for
alkenes, 15% for aromatics, and 15% for OVOCs, respectively. These approximate values vary
somewhat from city to city. At night, during summer, the major contributions in urban areas are
the oxidation by $NO_3$ (60% in Beijing, 45% in Chengdu, 15% in Shanghai and Guangzhou) and
ozone (20% in Beijing, 35% in Shanghai, and 25% in Chengdu). At the very remote station of
Waliguan, the relative daytime contributions to *AOC* in summer are 25% due to CO, 20% due to
methane, 40% due to OVOCs, and 10% due to alkenes. The corresponding contributions during
winter are 60% for CO, 20% for methane, 10% for OVOCs, and 5% for alkenes.

These model values can be compared, for example, with values calculated by Zhu et al. (2020)
from experimental data obtained in Shanghai. The peak value around noontime in summer is about
$(5-10) \times 10^7$ cm$^{-3}$ s$^{-1}$ in fair agreement with our estimates. In the wintertime, the values are between
$(5-8) \times 10^7$ cm$^{-3}$ s$^{-1}$, i.e., slightly higher than our calculated *AOC*. Feng et al. (2021) report a max-
imum value of about $1.7 \times 10^8$ cm$^{-3}$ s$^{-1}$ for the *AOC* at urban sites in Beijing during the summertime
of 2014. This number is close to our calculated value of *AOC* by $1.8 \times 10^8$ cm$^{-3}$ s$^{-1}$ in July. The peak
*AOC* value of $2.1 \times 10^7$ cm$^{-3}$ s$^{-1}$ reported by Liu et al. (2021) for the winter of 2018 in Beijing is
close to the model value of $3 \times 10^7$ cm$^{-3}$ s$^{-1}$.

### 7. Summary and Conclusions

The oxidizing capacity of the atmosphere can be characterized by different parameters including
the production and destruction rates of ozone and other oxidants, the ozone production efficiency,
the OH reactivity, and the length of the reaction chain responsible for the formation of ozone and
$RO_x$. The value of these parameters depends on whether ozone formation is limited by the availa-
bility of $NO_x$ or VOCs. It is also affected by the aerosol burden in the atmosphere, specifically by
the rate at which heterogeneous chemical reactions take place in the atmosphere. In the present
study, we have used a regional chemical transport model with a detailed chemical scheme to quan-
tify these parameters in several chemical environments in China. Such studies should be helpful



in determining the factors that are responsible for the documented changes in the oxidation capacity of the atmosphere and hence the mean concentration of surface ozone in different regions of the country.

Our study shows that during winter, the formation of ozone in most of the eastern China Plain is VOC-limited. The ozone formation in remote western regions of the country, however, are $NO_x$-limited. In the south, an intermediate situation prevails, except in the Pearl River Delta area, where the formation of ozone is VOC-limited. In summer, ozone formation is $NO_x$-limited in most regions of China except in the urban areas of Beijing, Shanghai, Guangzhou, and Chengdu.

Our model calculations conducted for the summer season show that the largest contribution to the formation of $RO_x$ radicals in rural areas is due to the photolysis of ozone followed by the reaction between the electronically excited oxygen atom with water vapor. The second largest contribution is provided by the photolysis of formaldehyde. In urban and suburban areas, the formation of $RO_x$ starts in the early morning with the photolysis of HONO, followed by the photolysis of HCHO and other OVOCs and finally of ozone. In polluted areas, the contribution of oxygenated VOCs is important and needs to be included in any oxidant budget analysis.

The summertime destruction of $RO_x$ radicals in the rural regions is principally due to radical-radical reactions, including $HO_2 + HO_2$, $HO_2 + RO_2$, and $HO_2 + OH$. In urban and suburban areas, the main destruction processes are associated with reactions between $NO_2$ and OH, and between $RO_2$ and NO. The destruction of radicals associated with the uptake of $HO_2$ on aerosol surfaces plays a limited role.

At all the considered/studied sites, the production rate of ozone is with a relatively equal contribution of the reactions of NO with $HO_2$ and $RO_2$, respectively. The source of $RO_2$ varies according to the region; it is mostly anthropogenic in urban areas and biogenic in remote areas. Values of ozone production rate are substantially higher in metropolitan areas than in remote areas: the maximum value in the early afternoon reaches 100 ppbv $h^{-1}$ in Beijing and Shanghai but is less than 2 ppbv $h^{-1}$ at the rural sites.

Our model simulations suggest that heterogeneous chemistry together with the effect of aerosol on light extinction (photolysis) contribute to an increase in the surface ozone concentrations by 4-6 ppbv during wintertime. In summer, the presence of the aerosol burden derived by our model leads to a reduction in surface ozone of up to 8 ppbv in the $NO_x$-limited areas of the central and eastern parts of China. The ozone concentrations, however, are enhanced by about 5-7 ppbv in the VOC-limited regions near Shanghai and Beijing. The reduction in the aerosol burden, resulting from the measures taken by Chinese authorities, will therefore lead to an increase of the $O_3$ density in the $NO_x$-limited rural areas and affect the efficiency of $O_3$ pollution control. A decrease of the $O_3$ concentration is expected in the VOC-limited areas. However, with the continuous reduction in



$NO_x$ emissions, the $NO_x$-limited areas tend to geographically expand, and an $O_3$ increase should therefore occur in a broader area.


The daytime averaged OH reactivity due to $NO_x$ varies from less than 1 s$^{-1}$ in the western part of the country to 10 s$^{-1}$ in the North China Plain during winter. It is closer to 3 s$^{-1}$ during summer except in large urban areas where it is close to 10 s$^{-1}$. The reactivity due to VOCs is very small in the western regions of the country, but varies from 2 to 10 s$^{-1}$ in eastern China, with somewhat
higher values in winter. The $VOC^R$-to-$NO_x^R$ ratio is higher than 5 over the Tibetan Plateau, but less than 1.5 in the eastern regions of the country and even smaller in $NO_x$-rich regions of the most polluted areas.

The number of cycles affecting $RO_x$ radicals during daytime before they undergo a termination
process is around 3 to 5 in remote areas. In January, it reaches as high as 8 to 10 cycles in the eastern plain and in the southern area of China. The values are low in northern and western China, where the NO concentration is low during winter. In the metropolitan areas during July, about 10 cycles are performed before the radicals are lost.

Finally, our model simulations suggest that the daytime oxidizing capacity is mostly influenced by the reaction of the OH radical with alkenes, carbon monoxide, and oxygenated VOCs, and to a lesser extent with aromatics. The relative contribution of different chemical processes varies with locations (urban versus rural) and with seasons (winter versus summer). During nighttime, the largest contributions are due to the oxidation of hydrocarbons by $NO_3$ and ozone.


With the reduction of $NO_x$ emissions observed in China, explicit consideration of nocturnal oxidative chemistry taking into account, the effect of nitrate radicals will become increasingly crucial for the assessment of air quality. With the reduction in the anthropogenic emissions of VOCs in China, the role of natural VOCs with high reactivity, such as isoprene, will become increasingly
important regarding oxidative processes, especially in scenarios with increasing temperature and extreme weather associated with climate change. To understand the contribution of photodegradable OVOCs to the oxidative capacity of the atmosphere and to the formation of secondary pollutants, additional studies that include systematic measurements of OVOCs and more accurate time-dependent estimates of emissions will be needed.


*Code and data availability.* The WRF-Chem model is public available at https://www2.mmm.ucar.edu/wrf/users/. The air quality data at surface station are public available
at the website of Ministry of Ecology and Environment of the People's republic of China at http://english.mee.gov.cn/.



*Author contributions.* JD and GB designed the structure of the manuscript, performed the numerical experiments, analyzed the results, and wrote the manuscript. JD analyzed the data and established the figures. All co-authors provided comments and reviewed the manuscript.

*Competing interests.* The authors declare that they have no conflict of interest.

*Acknowledgments.* The present joint Sino-German study was supported by the German Research Foundation (Deutsche Forschungsgemeinschaft DFG) and the National Science Foundation of China (NSFC) under the Air-Changes Project number 4487-20203. The National Center for Atmospheric Research (NCAR) is sponsored by the US National Science Foundation. We would like to acknowledge the high-performance computing support from NCAR Cheyenne.

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






Table 1. Simplified chemical mechanisms used for the interpretation of model results

| Reaction Number | Reactions | Reaction rate constant |
|---|---|---|
| Photolysis reactions | | |
| R1 | $O_3 + h\nu \rightarrow O(^1D) + O_2$ | (Emmons et al., 2010) |
| R2 | $NO_2 + h\nu \rightarrow NO + O$ | |
| R3 | $HCHO + h\nu + O_2 \rightarrow 2\ HO_2 + CO$ | |
| R4 | $OVOC_i + h\nu \rightarrow RO_{x,i}$ | |
| R5 | $HONO + h\nu \rightarrow NO + OH$ | |
| Thermal reactions | | |
| R6 | $O(^1D) + M \rightarrow O(^3P) + M$ | (Emmons et al., 2010) |
| R7 | $O + O_2 + M \rightarrow O_3 + M$ | |
| R8 | $O(^1D) + H_2O \rightarrow 2\ OH$ | |
| R9 | $CH_4 + OH + O_2 \rightarrow RO_2\ (= CH_3O_2) + H_2O$ | |
| R10 | $HC_i + OH + O_2 \rightarrow \alpha HO_2 + \beta RO_{2,i} + \text{oxidation products}$ | |
| R11 | $Alk_i + O_3 \rightarrow OH, HO_2, RO_{2,i}$ | |
| R12 | $CO + OH + O_2 \rightarrow CO_2 + HO_2$ | |
| R13 | $OH + O_3 \rightarrow HO_2 + O_2$ | |
| R14 | $HO_2 + O_3 \rightarrow OH + 2\ O_2$ | |



| | | |
|---|---|---|
| R15 | $OH + HO_2 \rightarrow H_2O + O_2$ | |
| R16 | $HO_2 + HO_2 + M \rightarrow H_2O_2 + O_2 + M$ | |
| R17 | $HO_2 + RO_{2,i} \rightarrow ROOH_i + O_2$ | |
| R18 | $RO_{2,i} + RO_{2,j} \rightarrow products$ | |
| R19 | $NO + O_3 \rightarrow NO_2 + O_2$ | |
| R20 | $NO + HO_2 \rightarrow NO_2 + OH$ | |
| R21 | $NO + RO_{2,i} + O_2 \rightarrow carbonyl + NO_2 + HO_2$ | |
| R22 | $NO + RO_{2,i} \rightarrow RONO_{2,i}$ | |
| R23 | $NO_2 + OH + M \rightarrow HNO_3 + M$ | |
| R24 | $NO + OH \rightarrow HONO$ | |
| R25 | $NO + NO_2 + H_2O \rightarrow 2\ HONO$ | |
| R26 | $HONO + HONO \rightarrow NO + NO_2 + H_2O$ | (Zhang et al., 2021) |
| R27 | $HONO + OH \rightarrow NO_2 + H_2O$ | |
| Heterogeneous reactions | | |
| R28 | $HO_2 \rightarrow H_2O$ | (Gaubert et al., 2020) |
| R29 | $N_2O_5 \rightarrow 2\ HNO_3$ | (Bertram et al., 2009; Yu et al., 2020) |
| R30 | $NO_3 \rightarrow HNO_3$ | (Liu and Wang., 2020) |



| R31 | NO$_2$ → 0.5 HONO + 0.5 HNO$_3$ | (Fu et al., 2019; Zhang et al., 2021) |
|------|------------------------------|------------------------------------|











Table 2. Location of the observational sites or areas referred to in the present study

| Site name | Latitude | Longitude | Site type |
|---|---|---|---|
| Beijing | 39.90° N-40.10° N | 116.30° E-116.50° E | Urban sites |
| Shanghai | 31.10° N-31.30° N | 121.40° E-121.60° E | Urban sites |
| Guangzhou | 23.00° N-23.20° N | 113.10° E-113.30° E | Urban sites |
| Chengdu | 30.50° N-30.70° N | 103.90° E-104.10° E | Urban sites |
| Wangdu | 38.66° N | 115.25° E | Rural site |
| Heshan | 22.73°N | 112.93°E | Rural site |
| Waliguan | 36.17° N | 100.54° E | Mountainous/Background site |
| Hok Tsui | 22.22° N | 114.25° E | Coastal/Background site |









Table 3. Sensitivity experiments

| Modeling cases | Description |
|---|---|
| *Het-All* | With all heterogeneous reactions and all other HONO sources |
| *No-HetHO2-Aero* | Without $HO_2$ uptake on aerosols |
| *No-HetNO3-Aero* | Without $NO_3$ uptake on aerosols |
| *No-HetN2O5-Aero* | Without $N_2O_5$ uptake on aerosols |
| *No-HetNO2-Aero* | Without $NO_2$ uptake on aerosols |
| *No-Het-Aero* | Without $HO_2$, $NO_3$, $N_2O_5$, and $NO_2$ uptake reactions on aerosols |
| *No-Phot* | Without aerosol effects on light extinction and photodissociation |
| *No-Het-Aero-Phot* | Without $HO_2$, $NO_3$, $N_2O_5$, and $NO_2$ uptake reactions on aerosols and aerosol effects on light extinction and photodissociation |
| *No-HONO* | Without HONO sources |
| *No-Het-HONO-Phot* | Without $HO_2$, $NO_3$, $N_2O_5$, and $NO_2$ uptake reactions on aerosols, other HONO sources and aerosol effects on light extinction and photodissociation |







Table 4. Comparison between values of the production rate of RO$_x$ [Units: ppbv h$^{-1}$] derived from local observations and calculated by our regional model in July 2018.

| Location | Period | Me.[a] | Ca.[b] | Me.[a] | Ca.[b] | Me.[a] | Ca.[b] | Me.[a] | Ca.[b] |
|---|---|---|---|---|---|---|---|---|---|
| | | Peak value[c] | | HONO[d] | | HCHO[e] | | Ozone[f] | |
| Beijing[g] | Spring 2018 | 7.0 | 7.5 | 4.0 | 4.0 | 1.5 | 2.0 | 1.5 | 1.5 |
| Shanghai[h] | Summer 2021 | 4.5 | 5.0 | 2.5 | 1.0 | 1.0 | 1.0 | 1.0 | 1.5 |
| Guangzhou[i] | Autumn 2018 | 5.6 | 6.0 | 2.0 | 3.0 | 1.5 | 1.0 | 1.0 | 1.5 |
| Chengdu[j] | Summer 2019 | 7.0 | 7.5 | 2.0 | 2.5 | 2.0 | 1.5 | 1.5 | 2.0 |
| Wangdu[k] | Summer 2014 | 5.0 | 4.8 | 2.0 | 2.0 | 1.0 | 2.0 | 1.5 | 1.0 |
| Heshan[l] | Autumn 2014 | 4.0 | 1.1 | 2.0 | 0.5 | 1.5 | 0.4 | 0.5 | 0.5 |

[a] Measured value in relevant periods; [b] Calculated value in our study in July 2018; [c] Peak value of production rate of RO$_x$; [d,e,f] Peak value of the production rate of RO$_x$ from the photolysis of [d] HONO, [e] HCHO and [f] O$_3$; [g,h,i,j,k,l] Observations from the studies of [g] Whalley et al. (2021), [h] Zhu et al. (2021), [i] Wang et al. (2022), [j] Yang et al., (2022), [k]Tan et al., (2017) and, [l] Tan et al. (2019).







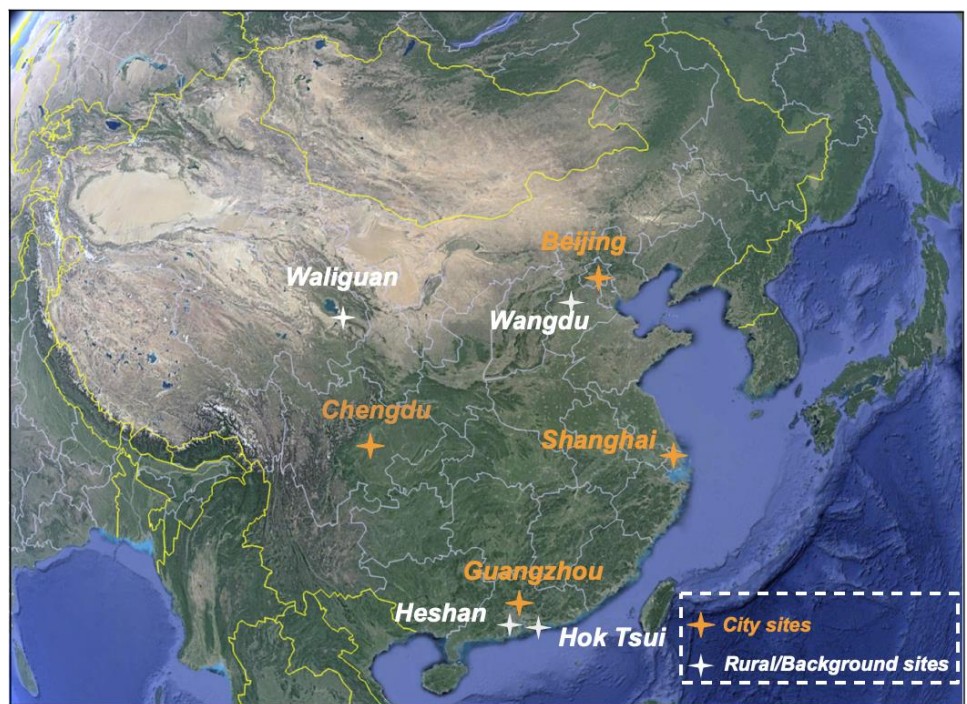

**Figure 1**. Location of sites (stars) considered in our analysis (from © Google Maps).







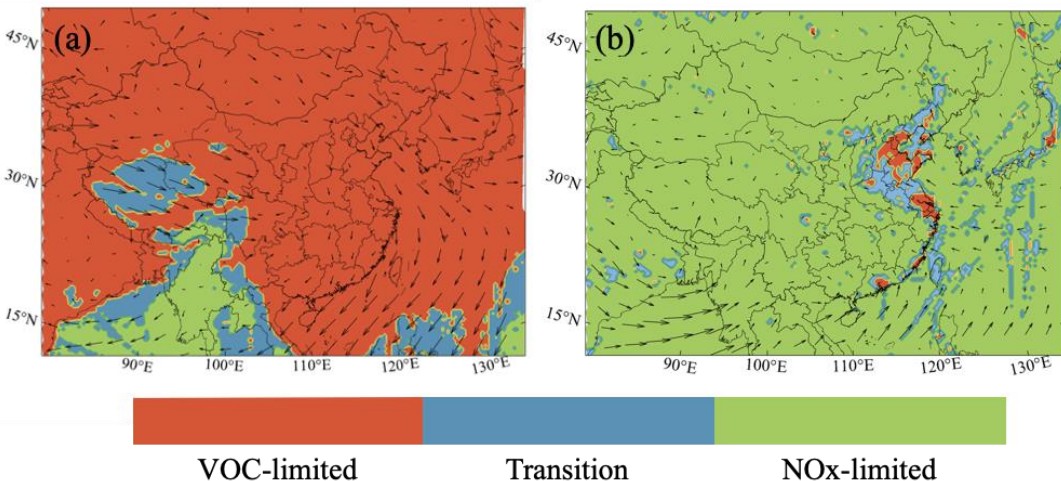

**Figure 2**. Display of regions in which the ozone production is limited by the availability of nitrogen oxides (NOx-limited, in green), and volatile organic carbon (VOC-limited, in red) in January (a) and July (b), 2018. The regions with intermediate conditions (Transition) are shown in blue. The indicator used to define these regions is the concentration ratio between formaldehyde (HCHO) and nitrogen dioxide (NO2).



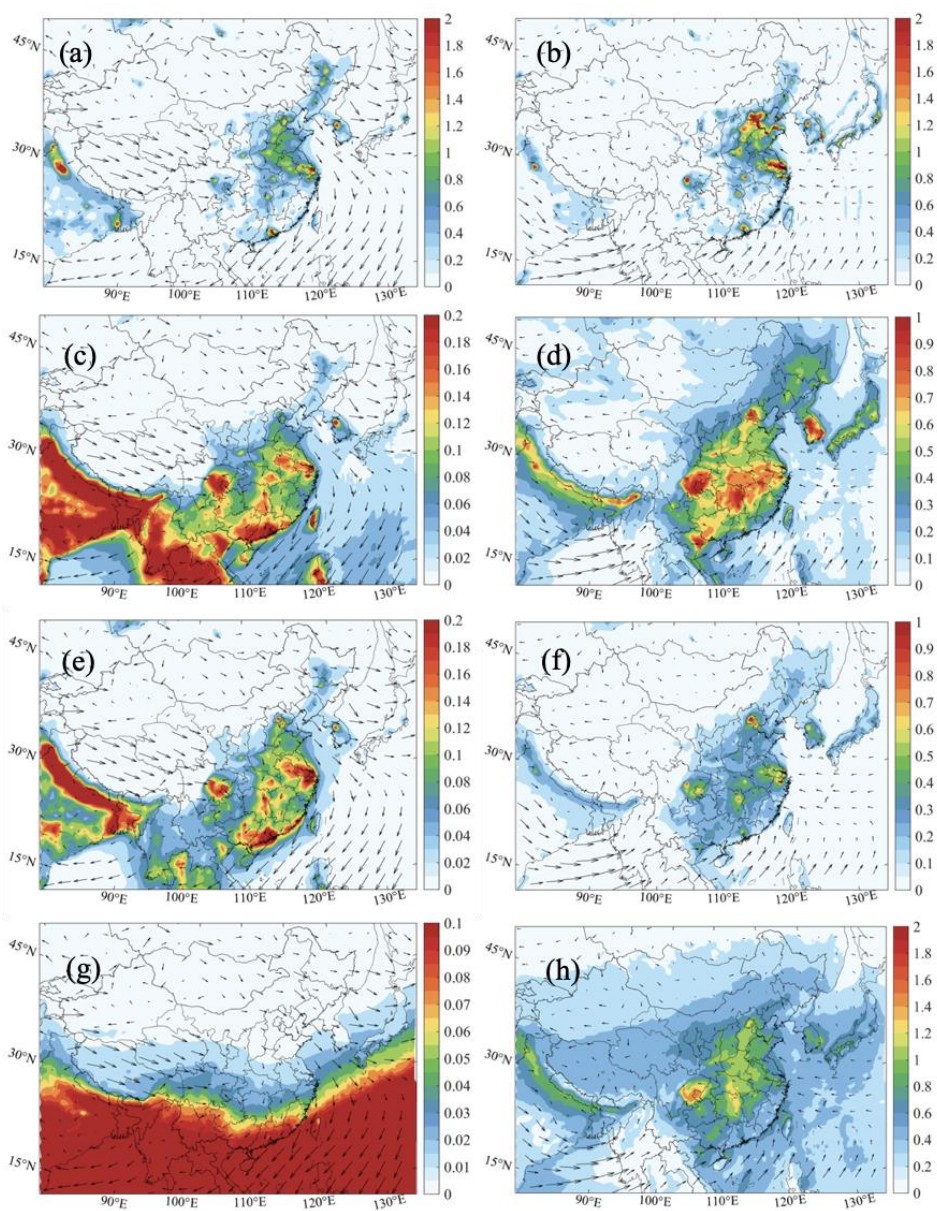


**Figure 3**. Spatial distribution of the production rate of $RO_x$ ($RO_2$+$HO_2$+OH) [$P(RO_x)$, Unit: ppbv h$^{-1}$] (*Het-All* case) from the photolysis of nitrous acid (HONO) (a, b), formaldehyde (HCHO) (c, d), non-HCHO oxidized volatile organic compounds (OVOCs) (g, h) and $O_3$ (Reactions between $O^1D$ and $H_2O$; e, f) in the daytime (08:00-19:00 Local Standard Time (LST)) of January (left column: a, c, e, g) and July (right column: b, d, f, h). Note the difference in scales among panels.






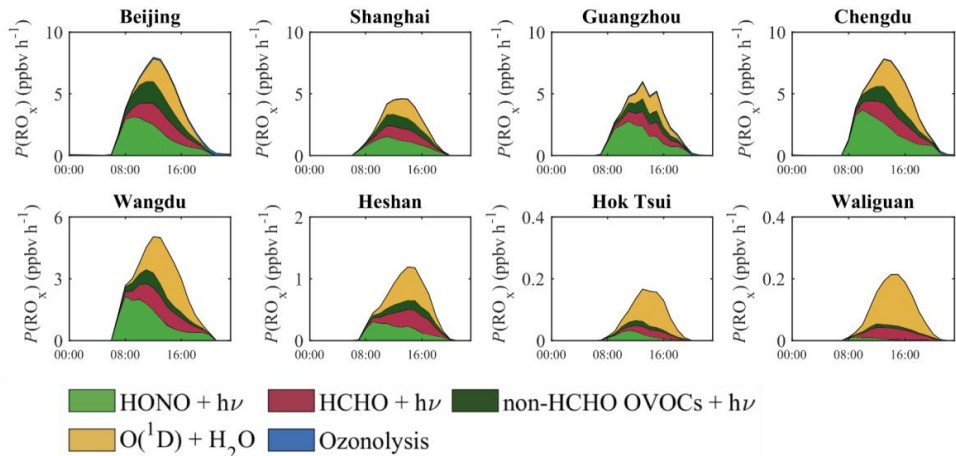

**Figure 4**. Diurnal variation of the production rate of $RO_x$ [$P(RO_x)$, Unit: ppbv h$^{-1}$] in different regions of China calculated for July 2018.





**Figure 5.** Spatial distribution of destruction rate of $RO_x$ [$D(RO_x)$, Unit: ppbv h$^{-1}$] (a, b) and the
relative contribution $L_N/D(RO_x)$ (c, d), $L_H/D(RO_x)$ (e, f) and $L_{het}/D(RO_x)$ (g, h) in the daytime of
January (left column: a, c, e, g) and July (right column: b, d, f, h).



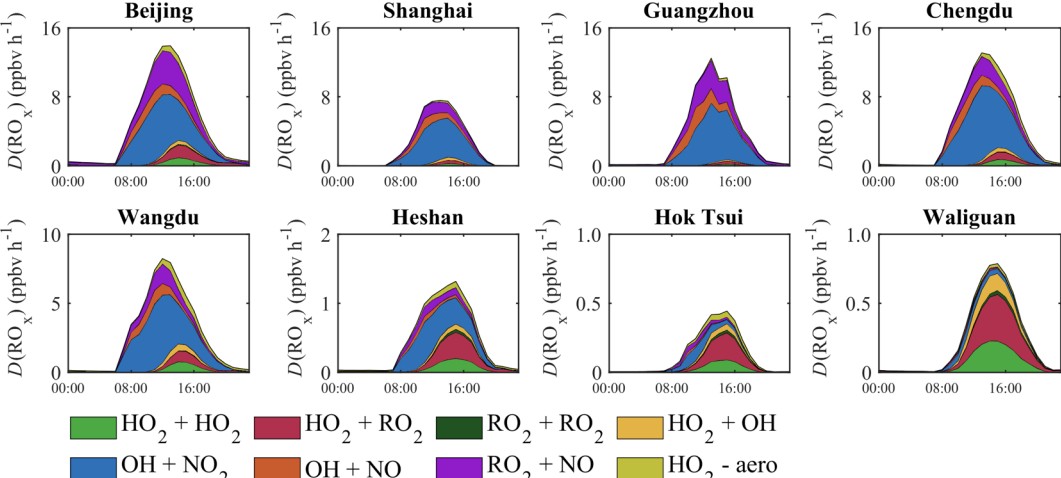

**Figure 6.** Diurnal variation of the photochemical destruction rate of $RO_x$ [$D(RO_x)$, Unit: ppbv h$^{-1}$] in eight sites of China for July 2018. The contributions to the destruction rate are the following: $L_H$ accounts for the following reactions: $HO_2 + HO_2$, $HO_2 + RO_2$, $RO_2 + RO_2$, and $OH + HO_2$. $L_N$ accounts for $OH + NO_2$ and $RO_2 + NO$. $L_{HO2}$ to the uptake of $HO_2$ by particles.



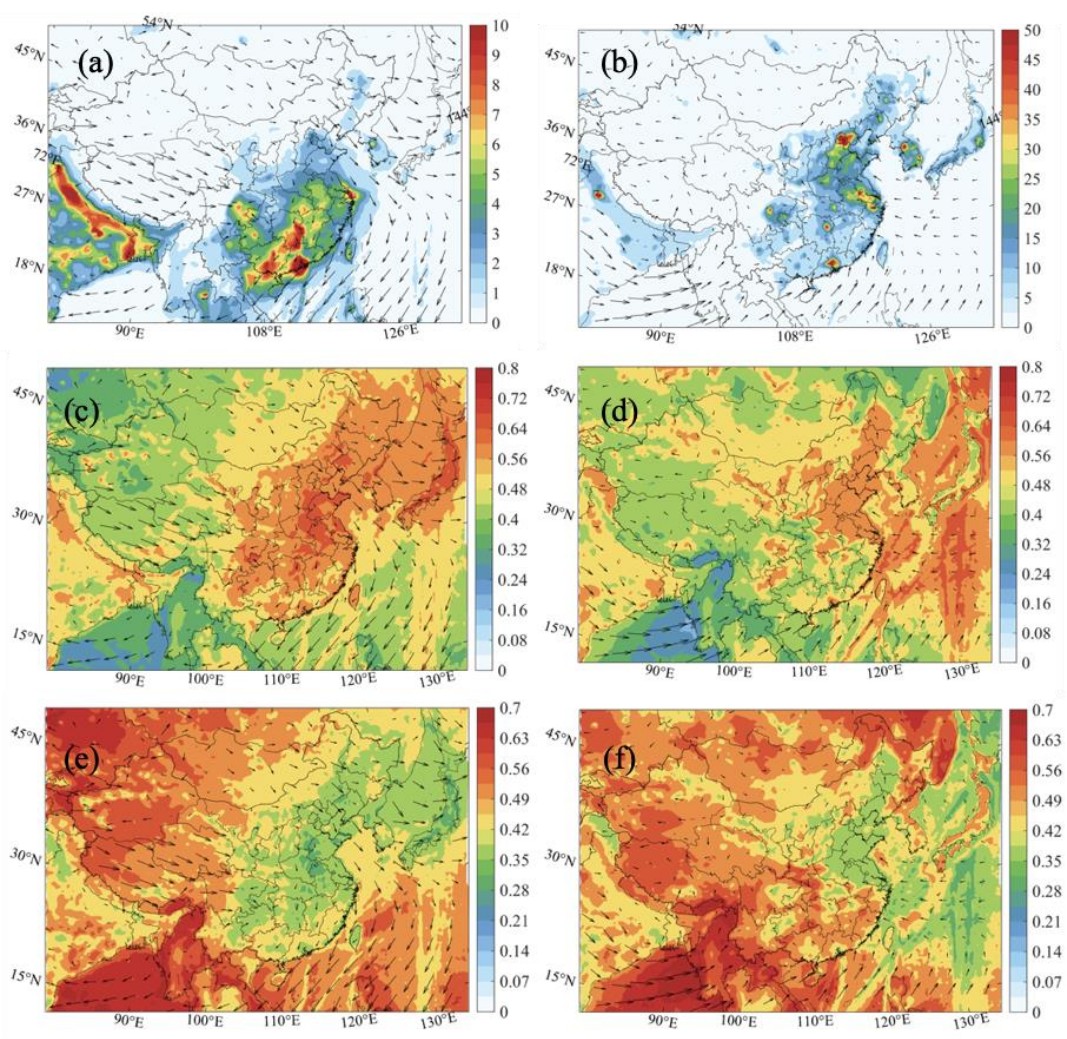

**Figure 7.** Spatial distribution of production rate of odd oxygen [$P(O_x)$, Unit: ppbv h$^{-1}$] (a, b) and the relative contributions from the reactions between HO$_2$ and NO (c, d) and RO$_2$ and NO (e, f) in the daytime of January (left column: a, c, e) and July (right column: b, d, f).







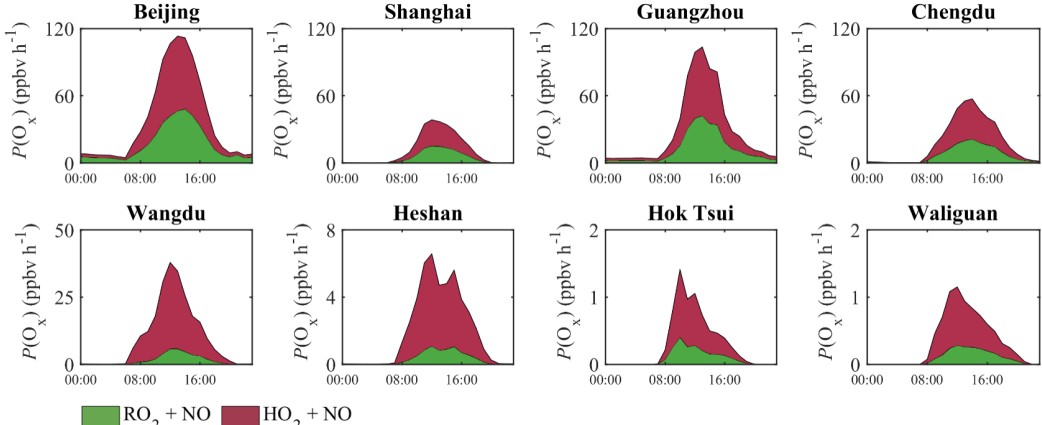

**Figure 8**. Diurnal variation of the $O_x$ production rate [$P(O_x)$, Unit: ppbv h$^{-1}$] in different regions of China for July 2018.



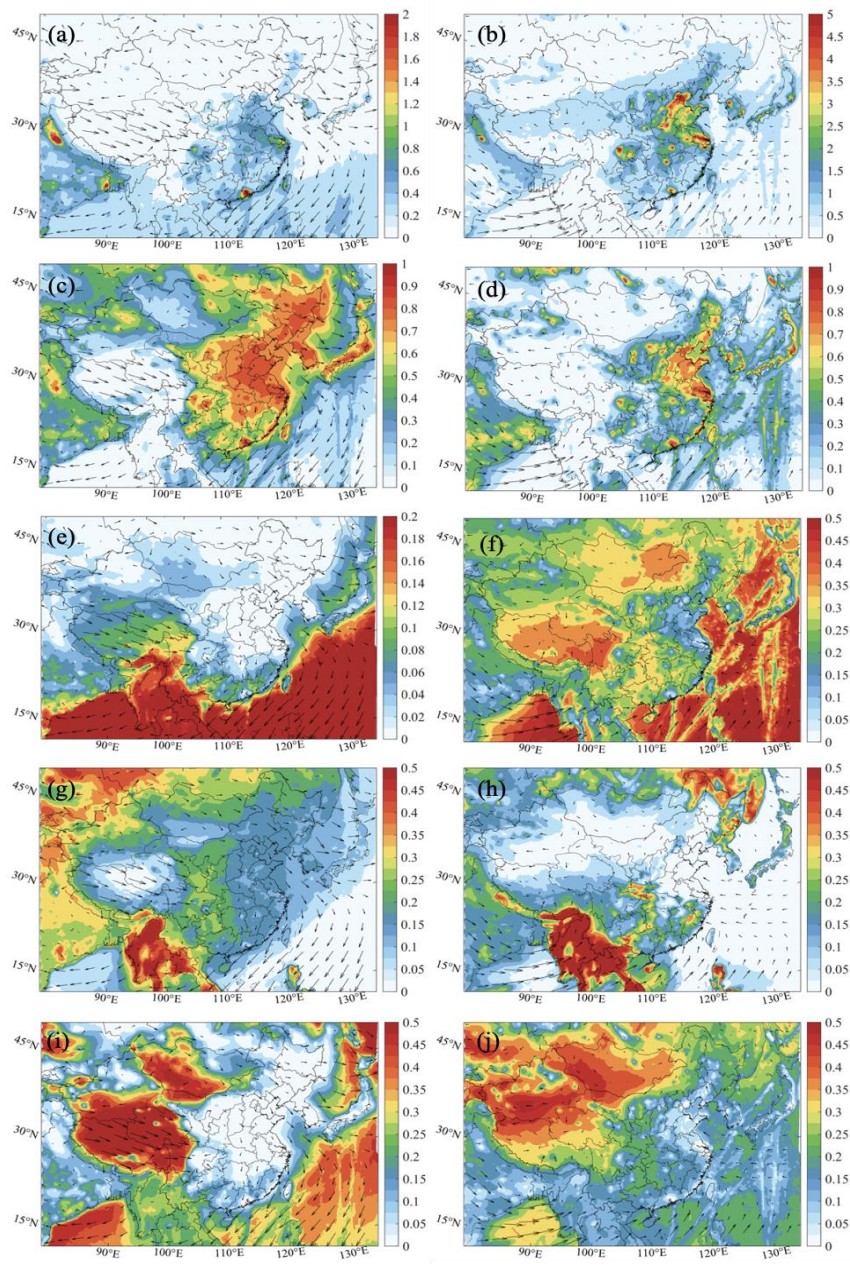


**Figure 9**. Spatial distribution of destruction rate of odd oxygen (a, b) [$D(O_x)$, Unit: ppbv h$^{-1}$] (*Het-All* case) and the relative contributions from the reactions of OH with $NO_2$ (c, d), $O(^1D)$ with $H_2O$ (e, f), Alkene with $O_3$ (g, h) and $HO_2$ with $O_3$ (i, j) in the daytime of January (left column: a, c, e, g, i) and July (right column: b, d, f, h, j).






**Figure 10.** Spatial distribution of the response of the monthly average surface $O_3$ concentration [Unit: ppbv] to the aerosol uptake by $HO_2$ (a, b; *Het-All* minus *No-HetHO2-Aero*), $N_2O_5$ (c, d, *Het-All* minus *No-HetN2O5-Aero*), $NO_2$ (e, f, *Het-All* minus *No-HetNO2-Aero*), and to the uptake by all these processes (g, h, *Het-All* minus *No-Het-Aero*). The results are shown for the daytime of January (left column: a, c, e, g) and July (right column: b, d, f, h) of 2018.

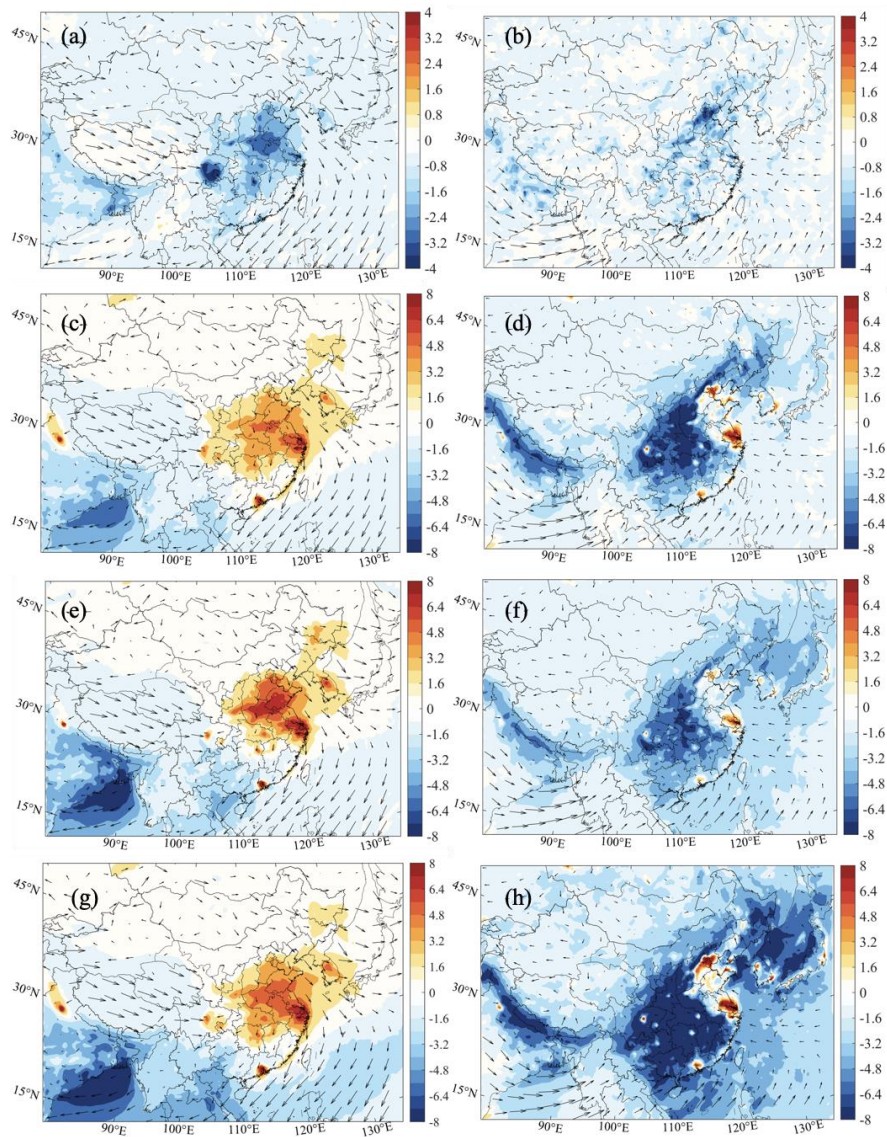

**Figure 11.** Changes in the surface concentrations of daytime $O_3$ [Unit: ppbv] resulting from the effect of aerosol-related solar light extinction on photolysis for January (a) and July(b) (*Het-All* minus *No-Phot*), the combined effect of photolysis and aerosol uptake for January (c) and July (d) (*Het-All* relative to *No-Het-Aero-Phot*), effects of $NO_2$ uptake by aerosols and on the surfaces as well as direct HONO emissions from traffic and gas phase formation for January (e) and July (f) (*Het-All* minus *No-HONO*), and from all $NO_2$, $N_2O_5$, $NO_3$ and $HO_2$ uptake processes photolysis effects and other HONO sources for January (g) and July (h) (*Het-All* minus *No-Het-HONO-Phot*).



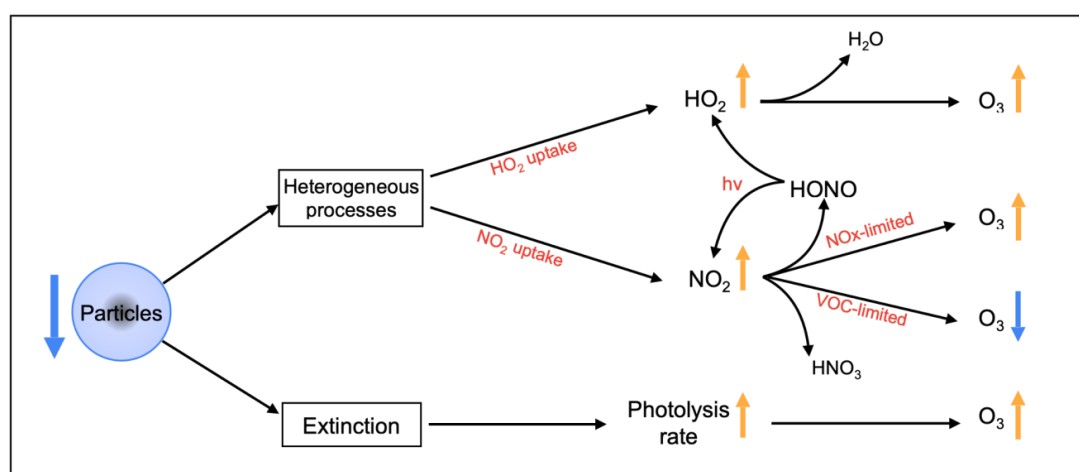


**Figure 12.** Schematic for the impact of aerosol through aerosol extinction of solar radiation and heterogeneous processes on ozone concentration. Arrows represent the changes in chemicals and photolysis rate associated with the reduction of aerosols.




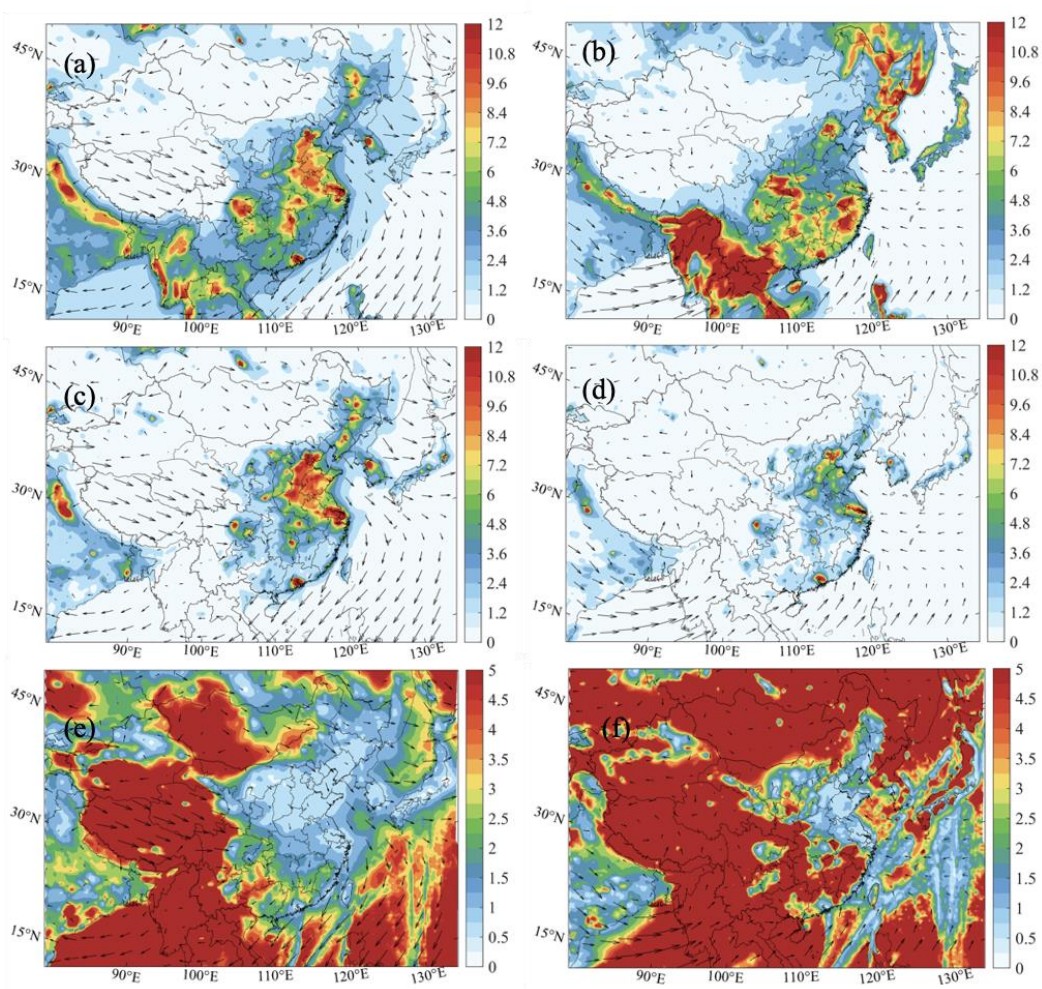


**Figure 13.** Spatial distribution of reactivity of VOCs ($VOC^R$) [Unit: s$^{-1}$] and NO$_x$ ($NO_x^R$; c, d) [Unit: s$^{-1}$], $VOC^R$ to $NO_x^R$ ratio (e, f) (*Het-All* case) for daytime conditions in January (left column: a, c, e) and July (right column: b, d, f).





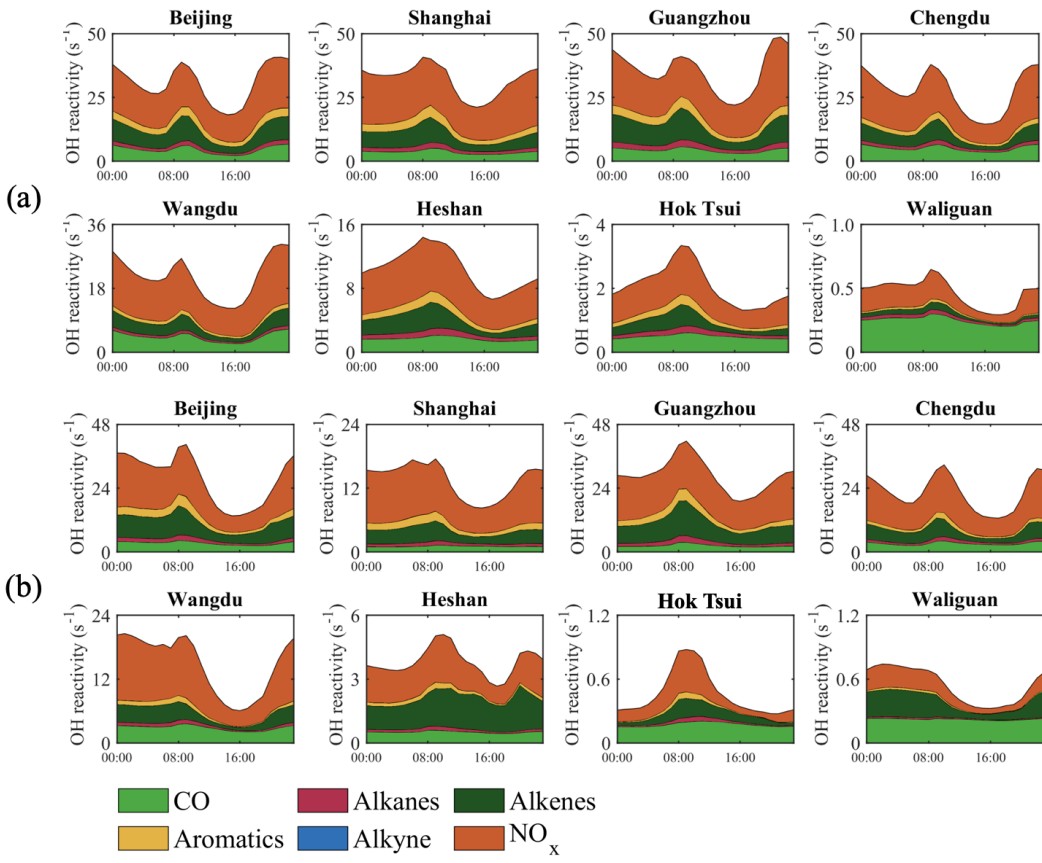


**Figure 14.** Diurnal variation of different contributions to the OH reactivity [Unit: s$^{-1}$] in cities and remote sites. The values associated with alkenes include the contribution of biogenic isoprene and terpenes. The two upper rows refer to January (a) and the two lower rows to July (b).





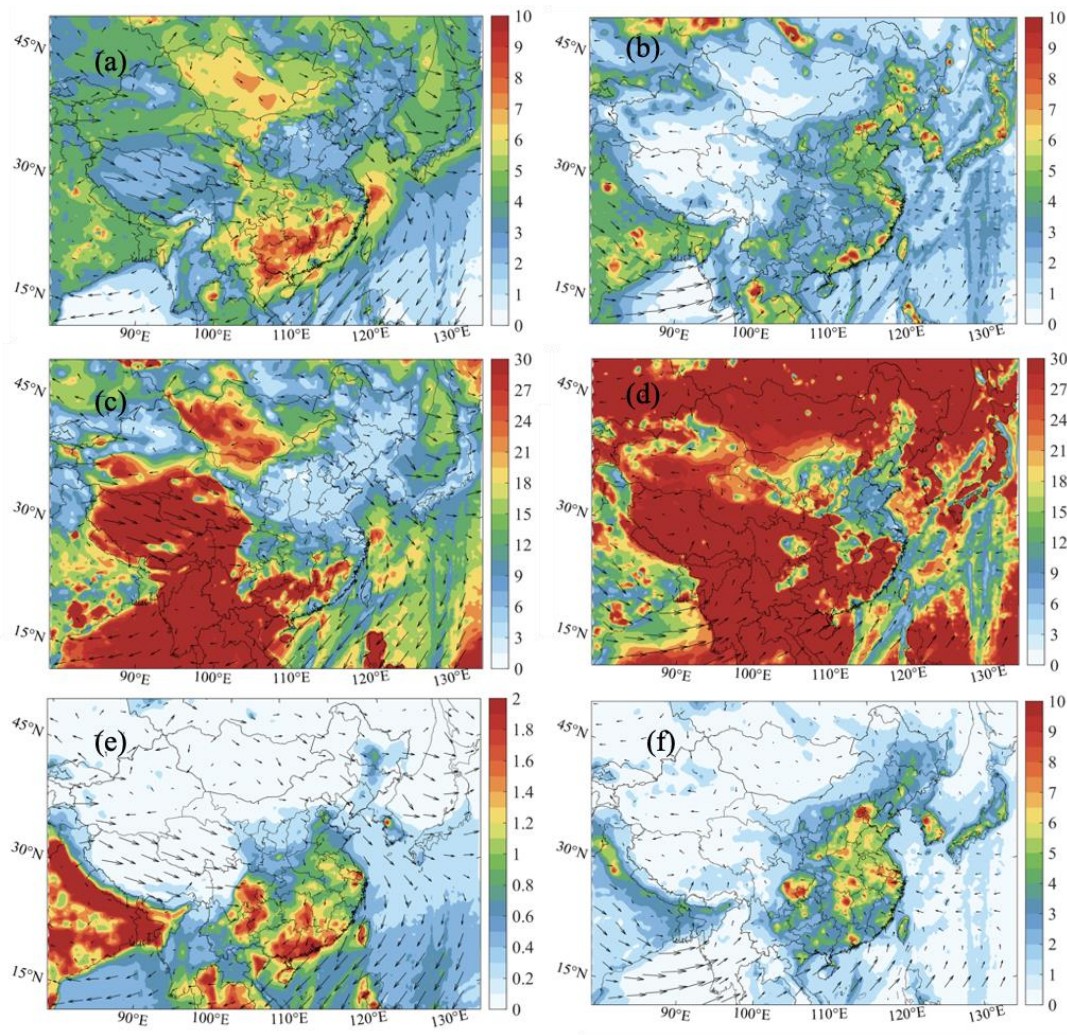


**Figure 15.** Spatial distribution of radical chain length (*ChL*) in January (a) and July(b), ozone production efficiency (*OPE*) in January (c) and July(d), and atmospheric oxidation capacity (*AOC* [Unit: $10^7$ molecular cm$^{-3}$ s$^{-1}$]) in January (e) and July(f) for daytime conditions extracted from the

*Het-All* case.




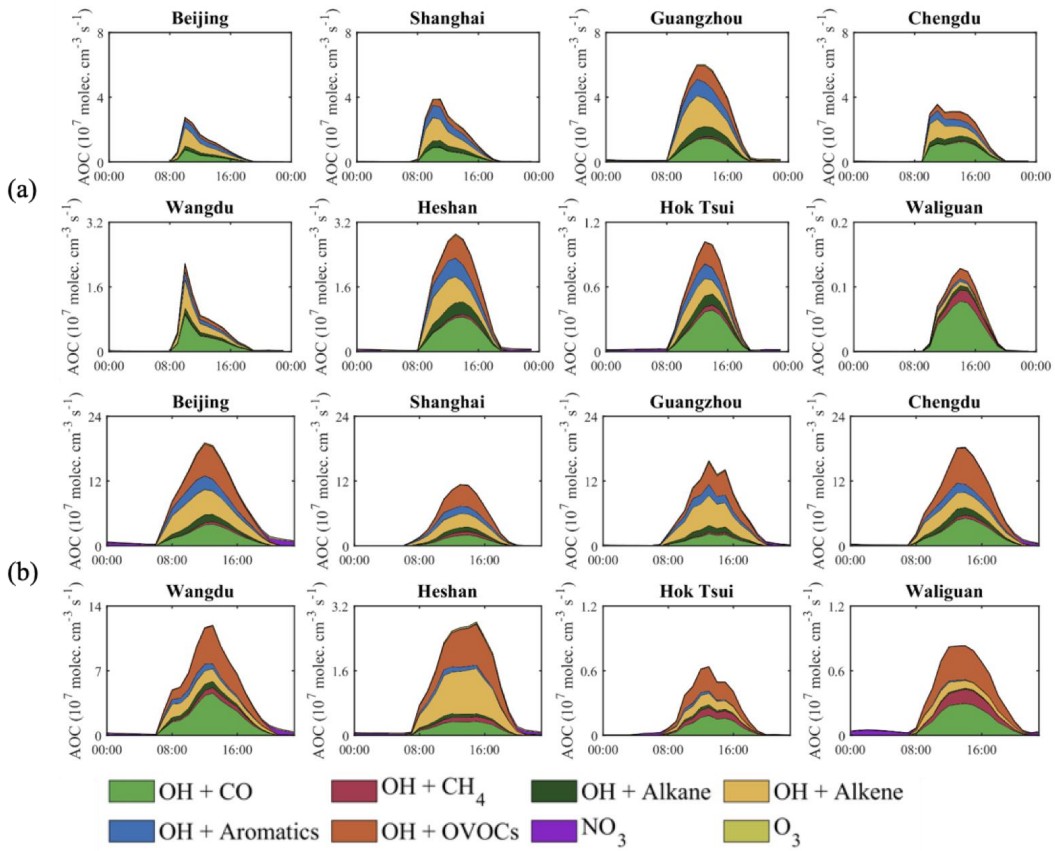

**Figure 16**. Diurnal variation of the atmospheric oxidizing capacity (*AOC*) [Unit: $10^7$ molecules cm$^{-3}$ s$^{-1}$] in cities and remote sites. The effect of alkenes includes the contribution of biogenic isoprene and terpenes, while the effect of OVOCs includes the contribution of formaldehyde. The two upper rows refer to January (a) and the two lower rows to July (b).



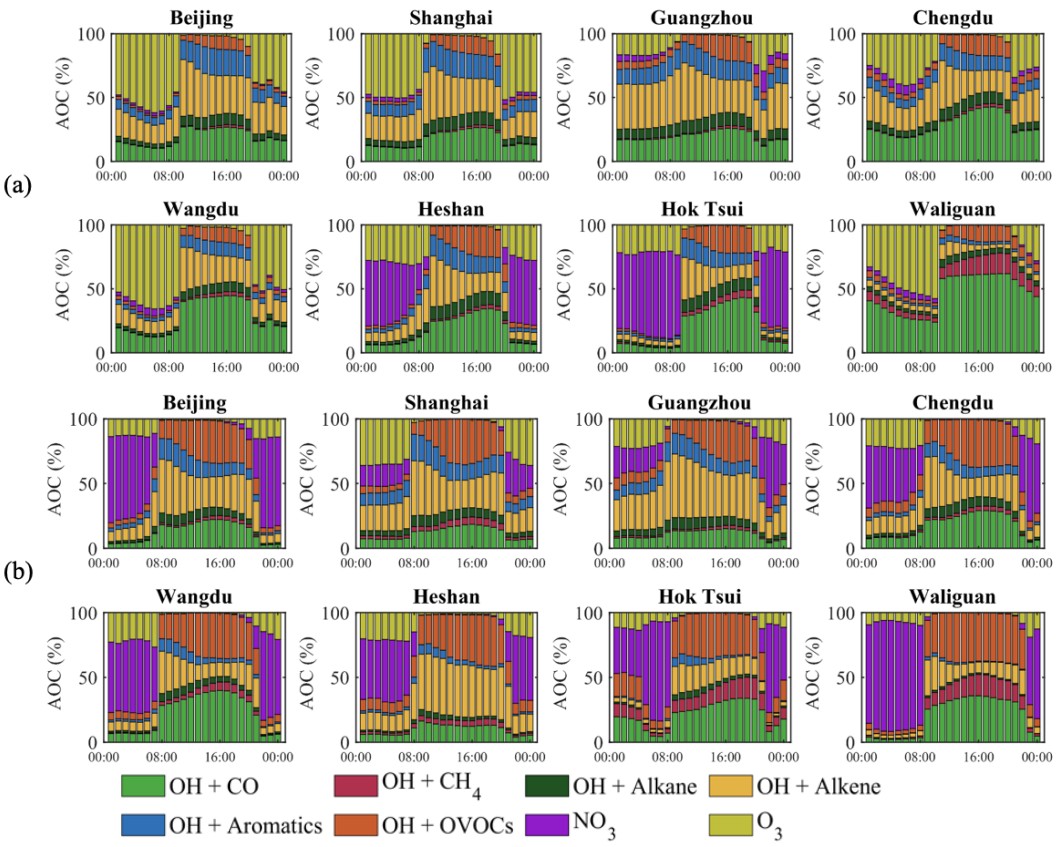


**Figure 17.** Same as Fig. 16, but expressed in relative terms [Unit: %] and highlighting the nighttime contribution to the *AOC* value. The two upper rows refer to January (a) and the two lower rows to July (b).