# Peer review of "The Atmospheric Oxidizing Capacity in China: Part 1. Roles of different photochemical processes"

_EGUsphere, 2023_

## Author Comment (AC1)

**Response to Reviewers' Comments**

**Dear Editor and Reviewers,**

Thank you very much for your efforts in handling and evaluating our submission.

The review comments are very helpful for improving the original manuscript. We have carefully considered and tried to address all of these comments in the revised manuscript. Below are the detailed point-by-point responses to the review comments. For clarity, the reviewer's comments are listed below in *black italics*, while our responses and changes in the manuscript are highlighted in blue and red, respectively.

We look forward to receiving a further evaluation of our work.

Best regards,

Guy Brasseur and co-authors

**Response to Reviewer #1:**

*The authors present a detailed narrative of WRF-Chem model outputs over China in one summer and one winter month during 2018. The main objective of this study is to characterize the current chemical conditions in China, particularly in light of the increasing ozone levels observed across the North China Plain since 2013. This manuscript provides a starting point for a companion paper that focuses on emissions changes.*

*While the manuscript does not necessarily present new science, the authors assess their model result with observations where possible, and provide many quantitative comparisons with prior studies. The topic is appropriate for ACP. The paper provides a useful and comprehensive quantitative assessment that the academic community will use as a useful point of comparison. I have a few comments below:*

**Response:** we thank the reviewer for the positive comments and constructive suggestions. We have addressed these comments and revised the original manuscript accordingly.

The validation of simulated VOCs and the analysis of model uncertainties in overestimated $PM_{2.5}$ and $NO_2$ was added to the revised manuscript. Below are the responses to specific comments.

**Major comments:**

*(1) I have some concerns about comparing outputs from a model at 36 km resolution to ground-based, urban observations. Does the coarse resolution cause any systematic biases?*

**Response:** The comparison between local ground-based urban observations with model output at a relatively large resolution is indeed a matter of concern. To alleviate this problem, we have combined the data from several stations inside a relatively coarse urban area, and have compared the average values. Although it is not a perfect approach, it is the only one we could use to obtain some insight into how the model performed. We have added a text in the manuscript to highlight this problem.

One should stress here that a comparison of coarse resolution model output with local measurements made at ground stations is not straightforward and can only provide crude information. In order to alleviate the problem, we have combined the concentration values measured by different stations within a given area with the 36 km resolution model results. The areas including the individual stations in metropolitan areas are provided in Table 2.

*(2) Model validation is lacking. The implications of model/observations discrepancies should be discussed. Specifically, There is no assessment for how well the model performs for VOCs.*

**Response:** The validation of the model for VOC is difficult for reasons stated in the text below. We have added the following text to the manuscript:

The validation of the model regarding volatile organic compounds is not easy to perform because of the short lifetime of most of these species, the inhomogeneity in their emissions, the complexity of the chemical processes involved, and the lack of observational data. In China, only a few stations report continuous measurements of VOCs. The comparison is made particularly difficult with a model whose grid size is equal to 36 km. Therefore, as an illustrative example, we show in Figure S13 of the Supplementary material, a comparison of the calculated and observed diurnal variation in the mixing ratio of ethane, propene, isoprene, ethane, propane, benzene, toluene, and xylene at the Hok Tsui site (Hong Kong) in January 2018.

*(3) Do PM2.5 overestimates in Beijing and elsewhere translate to the model overemphasizing the importance of heterogeneous processes? Could a model be generated with more accurate PM2.5 concentrations, or could the magnitude of the overestimate be further discussed when considering the metrics of choice?*

**Response:** The importance of heterogeneous processes is determined by the surface area density of the aerosol, which is affected by the concentration of the particles. There are no reliable measurements of surface area densities that we could use to validate our model. The concentration of $PM_{2.5}$ is certainly a factor that influences surface area density. The overestimation of the $PM_{2.5}$ concentration in large cities like Beijing is certainly a factor of uncertainty in the calculation of the heterogeneous conversion rates.

Based on our simulated results, in the $NO_x$-limited and Transition areas, the overestimation of aerosol concentration may cause an overestimation on the aerosol effect on ozone concentrations. We added a sentence discussing the possibility of the overestimated aerosol effect on ozone concentrations:

This value may be slightly overestimated in these regions since our calculated concentrations of aerosol are somewhat higher than the observed values.

The concentration of aerosol and $NO_x$ changed rapidly with time in China with consequences on the oxidizing capacity and heterogeneous processes. Part 2 of the paper is about the sensitivities of poorly represented processes including the aerosol load (in addition to the emissions of primary pollutants).

*(4) Similarly, NOx overestimates may complicate the analysis. If I understand correctly, an overestimate of NO2 changes dominant D(ROx) according to (line 679). The implications/ discussions of this are limited.*

**Response:** The calculated value of $D(RO_x)$ is dependent on the calculated concentrations of $HO_x$ and $NO_x$ species. It is difficult to determine the change of $D(RO_x)$ only to the overestimate of simulated $NO_2$. Therefore, we have added a sentence stating that

The calculated values of $D(RO_x)$ depend on the concentration values of the $NO_x$ and $HO_x$ radicals as provided by the model with the related uncertainties. The model overestimation of $NO_2$ reported in Section 3.3 may lead to an quantitative error in the contributions of different radicals to $D(RO_x)$ in other city sites (Guangzhou city).

*(5) The assessment of ozone production regimes through the use of formaldehyde to NOx ratios (FNRs) does not contribute to the discussion. FNRs are arguably useful when they are known to reflect more direct, mechanistic metrics such as LROX/LNOx. If the correlation is found/known/assumed, FNR observations can then be used to infer ozone production regimes. In this manuscript, no FNR observations are used, and direct metrics are already discussed. Therefore, the motivation for discussing FNRs is not well stated. Furthermore, there are documented issues with the use of "threshold" FNR values (see Souri et al. (2020) and subsequent papers). The citation provided for the threshold on line 522 (Jing et al., 2021) is missing from the list of references. Overall, I recommend that the authors either incorporate FNR observations, expand the discussion on what can be learned from this metric, or consider excluding the discussion leaving only the more mechanistic descriptors of ozone production regimes.*

**Response:** We agree with the suggestions. We now define the sensitivity regimes by the ratio between the $H_2O_2$ and $HNO_3$ production rates $[P(H_2O_2)/P(HNO_3)]$. The ozone sensitivity regimes are shown in Figure 2. An area is assumed to be VOC-limited or $NO_x$-limited if $P(H_2O_2)/P(HNO_3) < 0.06$ or $P(H_2O_2)/P(HNO_3) > 0.2$, respectively.

*(6) Figure S9: OH instead of HO on the y axis.*

**Response:** Changed

---

## Author Comment (AC2)

**Response to Reviewers' Comments**

**Dear Editor and Reviewers,**

Thank you very much for your efforts in handling and evaluating our submission.

The review comments are very helpful for improving the original manuscript. We have carefully considered and tried to address all of these comments in the revised manuscript. Below are the detailed point-by-point responses to the review comments. For clarity, the reviewer's comments are listed below in *black italics*, while our responses and changes in the manuscript are highlighted in blue and red, respectively.

We look forward to receiving a further evaluation of our work.

Best regards,

Guy Brasseur and co-authors

*The article titled "Characterizing Atmospheric Oxidation Capacity in China: Insights from Numerical Simulations" aims to provide a detailed analysis of the atmospheric oxidation capacity (AOC) in China, considering the changes in anthropogenic emissions over the past decade. The authors use the WRF-Chem model to simulate different parameters that influence AOC and investigate the impact of aerosols on surface ozone levels. The study also examines the contribution of various reactive species to AOC in different regions of China. The article provides a comprehensive analysis of AOC in China, considering various factors such as aerosol effects, photodissociation rates, and heterogeneous reactions.*

*The objectives of the study are clearly stated, and the article follows a logical structure, making it easy to understand the research approach and findings. Given the increasing concern about air pollution in China, understanding AOC and its contributing factors is highly relevant. The article addresses an important topic and sheds light on the impact of aerosols and anthropogenic precursors on atmospheric chemistry which is in the scope of ACP. I recommend the paper published in ACP after minor revision. Here is a few suggestions that might help to improve the paper.*

**Response:** We thank the reviewer for the positive comments and constructive suggestions. We have addressed all of these comments and revised the original manuscript accordingly. Below are the itemized responses to the specific comments.

*Discussion:*

*(1) The study heavily relies on numerical simulations performed with the WRF-Chem model. While this approach provides valuable insights, the authors should have discussed the limitations of the model and the uncertainties associated with the simulations more meticulously by adding a paragraph instead of shortly discussing in 3.3.*

**Response:** We added some discussion about model limitations in resolutions in simulating the radical and ozone chemistry. The uncertainties discussed are associated with the model performance in the concentration of $NO_2$, $PM_{2.5}$, and specific VOCs and the potential effect on ozone calculation. The new text for the model validation is as follows. It is complemented by text in the supplementary information

**3.3. Model validation**

[revised manuscript text omitted]

*(2) Table 1: Please provide the exact rate constants of each reaction.*

**Response:** We have added the reaction rate constants for each reaction in Table 1 of the paper with appropriate references.

*(3) Line 272: Distinct species have different gas-phase diffusion coefficients. 0.247 only applied to HO2. Please explain how the simulation's gas-phase diffusion coefficients worked.*

**Response:** In this study, we chose 0.247 to be the gas-phase diffusion coefficient for $HO_2$ aerosol uptake (Xue et al., 2016). For other gas-phase diffusion coefficients, we selected the value of 0.1, which is consistent with the modeling studies (Gaubert et al., 2020; Liu and Wang., 2020). We clarify our statement as:

$D$[cm² s$^{-1}$] is the gas-phase diffusion coefficient, with the value of 0.247 for $HO_2$ uptake (Mozurkewich et al., 1987; Xue et al., 2016) and 0.1 for $NO_2$, $NO_3$ and $N_2O_5$ uptake (Gaubert et al., 2020; Liu and Wang., 2020).

Xue, L., Gu, R., Wang, T., Wang, X., Saunders, S., Blake, D., Louie, P. K. K., Luk, C. W. Y., Simpson, I., Xu, Z., Wang, Z., Gao, Y., Lee, S., Mellouki, A., and Wang, W.: Oxidative capacity and radical chemistry in the polluted atmosphere of Hong Kong and Pearl River Delta region: analysis of a severe photochemical smog episode, Atmos. Chem. Phys., 16, 9891–9903, https://doi.org/10.5194/acp-16-9891-2016, 2016.

Gaubert, B., L. K. Emmons, K. Raeder., Correcting model biases of CO in East Asia: impact on oxidant distributions during KORUS-AQ, Atmos. Chem. Phys., 20, 14617-14647, https://doi.org/105194/acp-20-14617-2020, 2020.

Liu, Y. and Wang Tao, Worsening urban ozone pollution in China from 2013 to 2017 – Part 2: The effects of emission changes and implications for multi-pollutant control, Atmos. Chem.

Phys., 20, 6323-6337, https://doi.org/10.5194/acp-206323, 2020.

*(4) Line 280: The case of a large abundance of TMI in the aerosol liquid phase, which would result in the outcome of HO2 absorption being predominantly H2O, was put forth by Song et al. in 2021. The HO2 uptake products are not ambiguous. Please refer to Mao et.al, 2013 and Mao et.al, 2017 for more information on the HO2 uptake product as Mao et.al, 2013 has studied the potential reaction routes in detail. A test on how HO2 uptake produces H2O2 should be added.*

**Response:** To test the different impacts of $HO_2$ uptake on $H_2O$ and $H_2O_2$, two additional model cases with the $HO_2$ uptake producing $H_2O_2$ were performed for conditions corresponding to January and July 2018. The differences in the calculated surface mixing ratio of OH, $HO_2$, $H_2O_2$, and ozone for the two different assumptions are shown in Figure S21. When the $HO_2$ uptake produces $H_2O_2$, an increase in $H_2O_2$ concentration is produced by the model in both January (Figure S21e) and July (Figure S21f) of 2018. Consistently, the simulated concentration of $HO_2$ is enhanced (Figure S21 c, d), resulting from the photolysis of $H_2O_2$. However, there are no clear and consistent changes in the concentration of the OH radical (Figure S21 a, b) and of ozone (Figure S21 g, h). We added the following text:

Finally, we assess how the assumption made on the product of the $HO_2$ uptake influences our model results. Figure S21 in the Supplementary Information shows the differences in calculated near-ground mixing ratios of OH, $HO_2$, $H_2O_2$, and ozone when the heterogenous conversion of $HO_2$ is assumed to produce hydrogen peroxide rather than water molecules.

Two references (Mao et al., 2013; Mao et al., 2017) and the difference between the two different assumptions were added to the manuscript and the supplementary information (In Figure S21).

Mao, J., Fan, S., Jacob, D. J., Travis, K. R. Radical loss in the atmosphere from Cu-Fe redox coupling in aerosols. Atmospheric Chemistry and Physics, 13(2), 509-519. https://doi.org/10.5194/acp-13-509-2013, 2013.

Mao, J., Fan, S., Travis, K. R., Horowitz, L. W. Soluble Fe in aerosols sustained by gaseous $HO_2$ uptake. Environmental Science & Technology Letters, 4(3), 98-104. https://doi.org/10.1021/acs.estlett.7b00017, 2017.

*(5) Technical comments. Line 49 : What does "presence of aerosols" mean? The atmosphere always contains aerosols. Please provide the range of aerosol concentrations that would either increase or decrease ozone concentration.*

**Response:** We modified the sentence as follows:

The model shows that the aerosol effects related to extinction and heterogeneous processes produce a decrease in surface ozone of approximately 8-10 ppbv in $NO_x$-limited rural areas and an increase of 5-10 ppbv in VOC-limited urban areas. In this later case, the ozone increase is noticeable for aerosol concentrations ranging from 20 to 45μg/m$^3$ in July 2018 (Figure S3 b).

*(6)Line 431: At night, the oxidizing capacity is due to the oxidation by NO3 and O3. Please add references.*

**Response:** Two references were added:

Brown, S. S., and Stutz, J. Nighttime radical observations and chemistry. *Chemical Society Reviews*, 41(19), 6405– 6447. https://doi.org/10.1039/C2CS35181A, 2012.

Ng, N. L., Brown, S. S., Archibald, A. T., Atlas, E., Cohen, R. C., Crowley, J. N. Nitrate radicals and biogenic volatile organic compounds: Oxidation, mechanisms, and organic aerosol. *Atmospheric Chemistry and Physics*, 17(3), 2103– 2162. https://doi.org/10.5194/acp-17-2103-2017, 2017.

*(7)Line 830: Due to its low latitude, Shenzhen experiences a meteorological summer season (monthly average temperatures above 22 °C) from April to October. The compensation mechanism was crucial, despite the fact that this particular citation could not support this claim.*

**Response:** We have added a sentence stating that such a compensating mechanism was highlighted by Qu et al. (2023). The sentence is

"Such a compensation mechanism was highlighted by Qu et al. (2023) based on their model study performed in the YRD region for different seasons".

Qu, Y., Wang, T., Yuan, C., Wu, H., Gao, L., Huang, C., Xie, M. The underlying mechanisms of PM2. 5 and O3 synergistic pollution in East China: Photochemical and heterogeneous interactions. Science of The Total Environment, 873, 162434. https://doi.org/10.1016/j.scitotenv.2023.162434, 2023.

*(8)Line 1054-1061:The article's primary indicator of interest was AOC. In other works, the author compared AOC data. Please be detailed in your analysis and explain how the data differs or is comparable to that in the article.*

**Response:** The calculation of *AOC* based on our simulated results is consistent with the definition adopted in the compared studies (Feng et al., 2021; Liu et al., 2022; Zhu et al., 2020;). Our calculated *AOC* is therefore comparable with the observed studies.

---

## Referee Report (RR1)

The authors have addressed all reviewer remarks. The manuscript has been updated appropriately. I recommend the revised manuscript to be accepted as-is.